# GENATATORS: AB INITIO GENE ANNOTATION WITH DNA LANGUAGE MODELS

## ABSTRACT

Inference of gene structure and location from genome sequences - known as de novo gene annotation - is a fundamental task in biological research. However, sequence grammar encoding gene structure is complex and poorly understood, often requiring costly transcriptomic data for accurate gene annotation. In this work, we revisit standard evaluation protocols, showing that commonly used per-token and per-sequence metrics fail to capture the challenges of real-world gene annotation. We introduce and theoretically justify new biologically grounded interval level metrics, along with benchmarking datasets that better capture annotation quality. We show that pretrained DNA language model (DNA LM) embeddings do not capture the features necessary for precise gene segmentation, and that task specific fine-tuning remains essential. We comprehensively evaluate the impact of model architecture, training strategy, receptive field size, dataset composition, and data augmentations on gene segmentation performance. We show that fine-tuned DNA LMs outperform existing annotation tools, generalizing across species separated by hundreds of millions of years from those seen during training, and providing segmentation of previously intractable non-coding transcripts and untranslated regions of protein-coding genes. Our results thus provide a foundation for new biological applications centered on accurate and scalable gene annotation.

## 1 INTRODUCTION

The rapid development of DNA sequencing technologies, such as third-generation sequencing and Hi-C, has led to an exponential growth in the availability of genome assemblies across the tree of life. This genomic data is invaluable for fundamental research, biotechnology, and biomedicine, but raw DNA sequences alone are insufficient for most applications. In order to interpret these data, genomes must be annotated, which allows the identification of functional elements. Gene annotation is the most important here, since it identifies genes and reveals their structural elements, which is critical for almost all downstream applications.

A gene is a continuous subsequence of genomic DNA that serves as the template for transcription, the process by which RNA molecules are synthesized from DNA. Genes are directional, and their direction is defined collinear with the direction of RNA synthesis. Therefore, genes can appear in forward or reverse orientation relative to the reference genome (Appendix A Figure A1A). In the genomes, approximately half of annotated genes are in the forward orientation and half in the reverse.

The two largest gene classes are messenger RNAs (mRNAs) and long non-coding RNAs (lncRNAs) — this paper focuses only on them. In the human genome, approximately 40.5% of genes are annotated as mRNAs and 35.2% as lncRNAs. mRNAs encode proteins and their sequence is segmented into exons and introns, with exons containing coding sequence (CDS) and untranslated regions (UTRs) at the 5′ and 3′ ends (Appendix A Figure A1B). Translation of the CDS provides the amino acid sequence of proteins, each amino acid encoded by three CDS letters (codon); thus, even a single nucleotide shift in an exon boundary can change all downstream codons. By contrast, lncRNAs lack CDS and do not produce proteins, but instead regulate diverse biological processes, including chromatin remodeling, immune response, viral defense, and cancer progression (Mattick et al., 2023; Sharma et al., 2024).

Annotating lncRNAs is a qualitatively different task compared to annotating mRNAs. Protein-coding genes can often be recognized from conserved protein-coding fragments while lncRNAs lack such

signals, evolve more rapidly, and are often expressed only in specific tissues, which makes their detection particularly challenging without additional evidence such as RNA-seq.

Untranslated regions of mRNAs are also essential to annotate. Although they are not translated into proteins, UTRs influence transcript stability, translation efficiency, and localization (Castillo-Hair et al., 2024). They may encode short functional peptides, and mutations in UTRs can be linked to human diseases (Filatova et al., 2023). Thus, a complete view of gene structure requires accurate recovery not only of coding exons but also of UTRs and non-coding genes.

Learning the sequence rules that govern transcription and protein synthesis should, in principle, enable prediction of gene structure directly from DNA sequence. Methods that attempt this are known as *ab initio* gene predictors, yet in practice they underperform approaches that incorporate supplementary evidence beyond the genome sequence (Scalzitti et al., 2020). Common sources of such evidence include gene annotations from closely related species and RNA-sequencing data from the target species (Raghavan et al., 2022). However, these resources are not consistently available across organisms or conditions, which sustains the demand for robust *ab initio* gene annotation methods that deliver high-quality results from sequence alone.

In this work, we address these gaps by applying DNA language models to gene segmentation and developing GENATATORs, a family of fine-tuned models specifically designed for *ab initio* annotation. Using biologically inspired metrics, justified by theoretical analysis and empirical validation, we demonstrate that pretrained DNA language model embeddings are insufficient for precise segmentation, making task-specific fine-tuning necessary. We then investigate how architecture, input context length, species diversity in training data, and augmentation strategies affect performance. Finally, we benchmark GENATATORs against existing methods and evaluate generalization on human and other species, showing that our models achieve state-of-the-art performance in gene segmentation due to capacity to uncover previously untrackable lncRNAs and UTRs of mRNA, while maintaining comparable accuracy to the best existing tools on segmentation restricted to mRNA CDS.

## 2 RELATED WORK

Early *ab initio* approaches relied on probabilistic models such as AUGUSTUS (Stanke et al., 2004), which is based on HMMs that hardcode biological rules of gene grammar. These models capture statistical patterns of protein-coding genes, including the presence of a start codon to initiate CDS, a stop codon to terminate it, absence of in-frame stops within the CDS, and canonical dinucleotides at splice junctions. Such models are effective for identifying protein-coding genes but fail to capture UTRs and lncRNAs (Scalzitti et al., 2020). To address these gaps, deep learning methods have been introduced to learn gene segmentation rules from DNA sequence. Helixer used CNNs for gene segmentation (Stiehler et al., 2020), and Tiberius integrated CNN layers with a differentiable HMM decoder, achieving state-of-the-art accuracy on protein-coding gene annotation (Gabriel et al., 2024). Although effective, these models remain constrained. Tiberius focuses on protein-coding genes without explicit modeling of UTRs or lncRNAs, and its CNN backbone is restricted to relatively short contexts (up to 10Kb) despite many human genes exceeding 30 Kb and spanning over 1 Mb.

Large DNA LMs have emerged as versatile backbones for genomic predictions (Schiff et al., 2024; Fishman et al., 2025; Dalla-Torre et al., 2024; Marchal, 2024; Brixi et al., 2025; Zhou et al., 2023). Based on transformer or SSM architectures, they can be pretrained on large genomic datasets. DNA LMs have matched or surpassed classical approaches across tasks such as splice-site prediction, promoter identification, and polyA signal detection. SegmentNT (de Almeida et al., 2025), a fine-tuned Nucleotide Transformer DNA LM (Dalla-Torre et al., 2024) with a U-Net head, is a nucleotide-resolution classifier that outputs probabilities for each gene element directly from DNA sequence. Authors of SegmentNT also introduced variants of this model pretrained on expression data — SegmentBorzoi and SegmentEnformer. However, as we demonstrate below, classification performance on individual gene elements does not reliably reflect the accuracy of full gene reconstruction. Consequently, the utility of these models for real-world biological applications remains unclear.

Recently, AlphaGenome has been introduced as a foundation model of the genome that predicts multiple modalities from sequence, including RNA-seq, chromatin accessibility, and splicing-related outputs (Avsec et al., 2025). In the splicing domain, it performs nucleotide-level classification of

donor and acceptor sites, prediction of splice-site usage, and quantitative splice-junction prediction. While not being a gene annotation system, such splicing predictions of the model are directly relevant to exon–intron boundary detection and therefore to transcript assembly.

Alongside these methods, several benchmarks have been proposed to assess gene annotation-related tasks. GUE (Zhou et al., 2023) includes splice-site prediction; however, it assigns a single label to 400 bp input sequences, which makes it biologically irrelevant: gene annotation requires single-nucleotide precision in detection of boundary between gene elements. BEND (Marin et al., 2023) instead operates at the nucleotide level, but it uses short input sequences, relies on metrics that are not biologically rigorous, and does not evaluate critical elements such as UTRs or lncRNA genes. A detailed comparison between benchmarks developed in this work, BEND, and GUE is provided in Appendix B.

Building on these observations, it is clear that systematic evaluations of modern DNA LMs for full gene segmentation are still missing. In particular, SSMs have not been comprehensively benchmarked, and among transformer-based models, only a single context-extension method (Peng et al., 2023) has been applied to process genes longer than the default receptive field. A unified benchmark is therefore needed to clarify how modern DNA LMs perform on gene segmentation, especially for lncRNAs and UTRs that remain inaccessible to most existing tools.

## 3 FORMAL DEFINITION OF THE PROBLEM AND METRICS

We formalize gene segmentation as a multiclass and multilabel nucleotide level classification task. The objective is to learn a function $f$ that maps an input representation $\boldsymbol{X} \in \mathbb{R}^{N_l \times H}$ to an output label matrix $\boldsymbol{L} \in \mathbb{R}^{N_l \times 5}$, where $H$ is the token embedding dimension, $N_l$ is the input length in nucleotides, and 5 is the number of target classes which are exon, intron, coding sequence (CDS), $5'$ untranslated region, and $3'$ untranslated region.

### 3.1 SEGMENTATION SCORING

Segmentation performance can be assessed using conventional classification metrics such as precision, recall, f1-score and PR-AUC computed per class at the nucleotide level. However, these metrics evaluate classification independently for each nucleotide and therefore may not capture biological dependencies between predictions. For instance, a misclassification of a single nucleotide within a megabase long gene has negligible impact on the overall metric, while the same error can alter the interpretation of all downstream sequence, since shifting a protein coding exon boundary by one nucleotide modifies all downstream trinucleotide blocks and yields a different amino acid sequence, a frame shift effect known in molecular biology.

To address this limitation, we use *interval level* segmentation scoring inspired by prior work (Scalzitti et al., 2020). In this approach, a target interval is a continuous sequence of nucleotides with identical ground truth class labels. A predicted interval is counted as a true positive only when it has complete reciprocal overlap with a ground truth interval, which means that the predicted and true intervals coincide.

Formally, let the ground truth class label sequence be $L = (l_1, l_2, \ldots, l_{N_l})$. An interval $I_m = [i, j]$ is assigned to class $K$ when $l_k = K$ for all $k \in [i, j]$. For each class $K$, let $\mathcal{I}_{\text{pred}}^K$ be the set of predicted intervals and let $\mathcal{I}_{\text{true}}^K$ be the set of ground truth intervals. We compute the following quantities. True positives are the number of predicted intervals that exactly match a ground truth interval. False positives are the number of predicted intervals without an exact match in $\mathcal{I}_{\text{true}}^K$. False negatives are the number of ground truth intervals that are not recovered in $\mathcal{I}_{\text{pred}}^K$.

The final interval level f1-score for class $K$ is

$$\text{F1}_{\text{interval}}^K = \frac{2\,\text{TP}}{2\,\text{TP} + \text{FP} + \text{FN}}. \tag{1}$$

This metric penalizes biologically important segmentation errors and provides a realistic assessment of model performance.

We also extend interval level scoring to evaluate overall accuracy of gene structure prediction (defined as *gene level* metric). In gene level scoring, a gene is counted as a true positive only when all of its

intervals are reconstructed correctly. Reference annotations may include multiple valid transcript structures for the same gene, known as transcription *isoforms*, which define different segmentations. To account for this ambiguity, we use a gene level rule that accepts a prediction as correct when the predicted interval set exactly matches the interval set of any annotated isoform of the target gene. The current models `Tiberius` and `AUGUSTUS` rely on hard coded parameters tailored to coding sequence identification, which makes them unable to detect exons that include untranslated regions. Therefore, for protein coding transcripts we report two gene level metrics, one where the complete exon structure is reconstructed and one where only the coding sequence part is reconstructed. We compute these metrics separately for exon mRNA and CDS mRNA. For non coding transcripts such as lncRNA, which have no CDS annotation, we compute gene level metrics using exon intervals only. To obtain an overall gene level score we sum the number of correctly predicted lncRNA genes by exon matching and the maximum of exon mRNA and CDS mRNA counts for protein coding genes, which allows a fair comparison across models

$$\text{Score}_{\text{gene}} = \text{TP}_{\text{exon-lncRNA}} + \max\Big(\text{TP}_{\text{exon-mRNA}}, \text{TP}_{\text{CDS-mRNA}}\Big). \tag{2}$$

In Appendix C.1, we present a theoretical analysis that derives how sensitivity of conventional PR-AUC and interval level metrics scales with boundary errors, justifying the need for the latter. This is followed by empirical evidence in Appendix C.2, where we demonstrate that relying on PR-AUC can lead to incorrect model rankings.

## 4 Experiments

**Input data** The training dataset consists of genes from all human chromosomes except 8, 20, and 21, which were held out for validation during training. When specified, we also included genes from all chromosomes of 39 additional mammalian species. All models were evaluated on human chromosome 20, since the human genome provides the most accurate annotation among all available species. For genes with multiple annotated isoforms, we selected a single isoform per gene with the longest cumulative length of exons. A detailed description of dataset preparation is provided in Appendix D.

**Models** We evaluated models representing different families of DNA LM architectures. From the SSM family, we included `Evo2-1B` (Brixi et al., 2025) and `Caduceus` (with PH and PS modifications) (Schiff et al., 2024). For Transformer-based models, we selected `GENA-LM` equipped with Recurrent Memory Transformer (RMT), capable of processing sequences comparable in length to complete genes (Kuratov et al., 2024). `DNABERT-2`, `DNABERT-S`, and similar architectures were not included due to the limited receptive fields. Additionally, we incorporated previously developed gene segmentation models based on the `Nucleotide Transformer` DNA LM: (`SegmentNT` and `SegmentNT_multispecies`), as well as models pretrained on gene expression data (`SegmentEnformer` and `SegmentBorzoi`), as well as classical models (HMM-based `AUGUSTUS` and the CNN&HMM hybrid `Tiberius`), in the final benchmarks. However, we did not evaluate embeddings, re-optimize dataset preparation or training procedures for these models, as such studies have been reported previously (de Almeida et al., 2025; Gabriel et al., 2024). We refer to the Appendix E Table A7 for the summary of all models benchmarked in this study.

For models operating at single-nucleotide resolution (`Evo2` and `Caduceus`), we appended a linear projection layer of shape $(H, 5)$ to map the model outputs to the five target classes. For non single-nucleotide resolution models (`Nucleotide Transformer`, `GENA-LM`), token embeddings were upsampled by repeating each token representation to match its corresponding nucleotide span and further processed using a U-NET architecture as proposed in de Almeida et al. (2025).

All models were trained using cross-entropy loss, and the best-performing checkpoint was selected based on exon-level f1-score on the validation set. Further details on model architectures and training protocols are provided in Appendix D.

### 4.1 Training on embeddings

DNA language models are expected to capture essential genomic features during pretraining. To evaluate whether gene-structure information can be extracted directly from frozen representations,

we conducted experiments where the DNA LM weights were fixed and only a shallow classifier was trained. Specifically, we used a linear projection layer for models operating at nucleotide resolution (`Evo2`, `Caduceus`) and a U-Net decoder for the token-based `GENA-LM` (byte–pair–encoded inputs).

As shown in Appendix F Table A8, none of the models produced embeddings containing sufficient information for accurate gene segmentation (see Appendix G, Table A9 for detailed metrics). The slightly higher performance of `GENA-LM` is likely attributable to the U-Net decoder, which, unlike the linear layer used in `Evo2` and `Caduceus`, can aggregate local contextual signals.

To understand why pretrained models fail at segmentation, we analyzed final-layer hidden states on ten randomly selected human genes (six mRNA and four lncRNA) using `Caduceus` and `GENA-LM`. For `GENA-LM`, which uses BPE tokens, we expanded each token embedding uniformly across its nucleotide span to obtain one vector per base for both models. PCA projections of the `Caduceus` embeddings revealed four distinct clusters corresponding to nucleotide identity (A, C, G, T), rather than gene structure (Appendix H, Fig. A4). `GENA-LM` embeddings formed diffuse clusters that also did not align with gene elements (Appendix H, Fig. A3). This contrasts sharply with embeddings obtained after fine-tuning on the gene segmentation task described in Section 4.2, which show clear separation of gene elements (Appendix H, Fig. A3). Quantitatively, fine-tuning increased the homogeneity of $k$-means ($k{=}5$) clusters with respect to exon, intron, CDS, 5′UTR, and 3′UTR labels from 0.003 to 0.583 for `Caduceus` and from 0.0 to 0.497 for `GENA-LM`.

These probing experiments should be interpreted in the context of our evaluation setup. In contrast to the BEND (Marin et al., 2023) benchmark, where authors employ task-specific trainable heads on top of frozen representations, our goal here is to assess what current pretrained encoders capture without relying on complex decoders. To keep the probing strictly aligned with this goal, we use only minimal heads (a linear layer for `Evo2` and `Caduceus`, and a shallow U-Net for `GENA-LM` to enable nucleotide resolution), which reveals the information present in the embeddings themselves rather than what can be recovered by a powerful decoder.

Together, these results indicate that pretraining alone is insufficient to encode the features required for precise gene segmentation and that task-specific fine-tuning remains essential for achieving high segmentation accuracy.

## 4.2 FINE-TUNING OF DNA LANGUAGE MODELS

We next conducted a series of fine-tuning experiments, where both the DNA LM parameters and the classification head were trainable. These experiments were designed to systematically investigate how model architecture and the biological information available during training influence gene segmentation performance.

As a baseline, we considered models trained on human genomic sequences with a model context length of 4,096 bp. Building on this setup, we explored the effect of extending the model context to 32 Kb, which provided a broader genomic window. We also examined whether expanding the training data to include genes from 39 additional mammalian species improved performance by leveraging evolutionary conservation, and we tested the impact of restricting the training set to protein-coding transcripts while excluding lncRNAs, so that the models were exposed only to sequences with well-defined coding structures. Finally, in a complementary experiment, we evaluated training on multiple isoforms per gene vs using single representative isoform per gene in the baseline. In all experiments we focused on `Caduceus PS` and `Caduceus PH` as representative SSMs, while `GENA-LM` served as the representative Transformer-based model, and we did not include `Evo-2`, since its larger size exceeded our available resources for running multiple fine-tuning experiments.

Our results (Table 1 and Appendix G Table A11) indicate that increasing the input sequence length yields the most substantial improvement in segmentation performance, with approximately 1.6–2× gains across models. Incorporating multiple species into the training set improved performance by approximately 1.2–1.5×. Excluding lncRNAs from the training data resulted in improved CDS detection for both `Caduceus` models. However, this came at the expense of reduced lncRNA segmentation performance, although the decrease was not as pronounced. This observation suggests that the sequence grammar underlying non-coding transcripts can, to large extent, be learned from protein-coding sequences. In contrast to CDS detection, we did not observe consistent improvements in exon segmentation for protein-coding genes when excluding lncRNA. Specifically, `GENA-LM` and

Table 1: Gene-level performace metrics for dataset and model modifications: absolute number of correctly reconstructed genes (abs) and differences (diff) compared to baseline (presented in the first column). The gene type (all+) means that all isoforms of all genes were included to the dataset.

| Chunk length ($N_l$), bp | | | 4096 | 4096 | | 32000 | | 4096 | | 4096 | | 4096 | | 4096 | | 4096 | |
|---|---|---|---|---|---|---|---|---|---|---|---|---|---|---|---|---|---|
| Gene type | | | all | all+ | | all | | mRNA | | all | | all | | all | | all | |
| Species | | | human | human | | human | | human | | 39 mammals | | human | | human | | human RC & splice-site filter | |
| Test-time augmentation | | | no | no | | no | | no | | no | | RC | | splice-site filter | | RC & splice-site filter | |
| **Model/dataset** | **Gene type** | **Class** | **abs** | **abs** | **diff** | **abs** | **diff** | **abs** | **diff** | **abs** | **diff** | **abs** | **diff** | **abs** | **diff** | **abs** | **diff** |
| GENA base | mRNA | exon | 31 | 22 | –9 | 61 | 30 | 33 | 2 | 45 | 14 | 41 | 10 | 42 | 9 | 57 | 26 |
| | | CDS | 1 | 0 | –1 | 1 | 0 | 1 | 0 | 5 | 4 | 4 | 3 | 1 | 0 | 6 | 5 |
| | lncRNA | exon | 15 | 10 | –5 | 24 | 9 | 11 | –4 | 18 | 3 | 20 | 5 | 24 | 13 | 29 | 14 |
| | all RNA | exon | 46 | 32 | -14 | 85 | 39 | 44 | –2 | 63 | 17 | 61 | 15 | 66 | 22 | 86 | 40 |
| Caduceus PH | mRNA | exon | 50 | 41 | –9 | 97 | 47 | 46 | –4 | 78 | 28 | 85 | 35 | 61 | 15 | 107 | 57 |
| | | CDS | 1 | 1 | 0 | 2 | 1 | 11 | 10 | 58 | 57 | 5 | 4 | 4 | –7 | 7 | 6 |
| | lncRNA | exon | 6 | 4 | –2 | 23 | 17 | 5 | –1 | 11 | 5 | 8 | 2 | 6 | 1 | 12 | 6 |
| | all RNA | exon | 56 | 41 | -15 | 120 | 64 | 51 | –5 | 89 | 33 | 93 | 37 | 67 | 16 | 119 | 63 |
| Caduceus PS | mRNA | exon | 68 | 43 | –25 | 112 | 44 | 76 | 8 | 91 | 23 | 101 | 33 | 77 | 1 | 126 | 58 |
| | | CDS | 20 | 0 | –20 | 6 | –14 | 32 | 12 | 94 | 74 | 24 | 4 | 23 | –9 | 30 | 10 |
| | lncRNA | exon | 9 | 2 | –7 | 18 | 9 | 4 | –5 | 4 | –5 | 17 | 13 | 9 | 0 | 18 | 9 |
| | all RNA | exon | 77 | 45 | -32 | 130 | 53 | 80 | 3 | 95 | 18 | 118 | 38 | 86 | 9 | 144 | 67 |

`Caduceus PS` achieved a modest improvement of approximately 10%, whereas `Caduceus PH` exhibited a similar decrease in performance. Overall, we concluded that transcript filtering does not substantially improve training performance. We also found that using multiple isoforms per gene slightly reduced accuracy, confirming that the single-isoform strategy remains preferable

To investigate the biological features underlying model errors, we analyzed the precision and recall of exon interval detection, stratifying exon-intron boundaries based on their flanking dinucleotide sequences (Appendix I, Fig. A5). Although the frequency of predicted boundaries at each dinucleotide generally reflects the true distribution, we identified samples where dinucleotides flanking predicted boundaries never occur at boundary positions in the actual data. Explicitly excluding exons flanked by these "illegal" dinucleotides, designated as a "splice site filter" improves performance of all models (Table 1).

As noted in the Introduction (Fig. A1A), genes occur in both orientations relative to the reference genome, and for this reason we apply a test-time reverse-complement (RC) augmentation in which each sequence is processed in its reference and RC orientations and the predictions are averaged. As shown in Table 1, this approach yields substantial improvements in performance for all models. Notably, `Caduceus PS`, whose architecture explicitly enforces RC equivariance in the DNA input representation, still benefits significantly from test-time RC augmentation and achieves a $\approx 1.5\times$ improvement in performance. This effect arises because sequences are segmented into fixed-size chunks and opposite orientations induce different chunkings, so averaging behaves like an ensembling. Furthermore, RC augmentation provides greater performance gains than applying a splice-site filter for both `Caduceus` models. To the best of our knowledge, this is the first study applying reverse-complement augmentation in the context of the gene segmentation task.

Finally, we compared performance across model architectures. Consistent with previous benchmarks (Schiff et al., 2024), `Caduceus PS` outperformed `Caduceus PH` in all experimental settings. The Transformer-based `GENA-LM` exhibited superior performance in lncRNA detection, whereas the SSM `Caduceus` detected a substantially higher number of protein-coding genes and achieved markedly better CDS segmentation compared to `GENA-LM`. We hypothesized that nucleotides counting is required to identify triplet-organized CDS. Whereas `GENA-LM` utilizes variable-length BPE tokens, making counting task challenging, `Caduceus` employs single-nucleotide tokenization, which may explain improved performance for the CDS class. In contrast, `GENA-LM` consistently outperformed `Caduceus` in lncRNA segmentation, a task that is more challenging than mRNA for both models, and this advantage aligns with model capacity, since `GENA base` has approximately 120M parameters compared to 16M in `Caduceus`. When we trained the same base

setup but with the larger 360M parameter `GENA-LM`, lncRNA segmentation performance improved by 25%, further highlighting the benefits of model scaling for this task (Appendix G, Table A10).

### 4.3 SCALING

To further improve model performance, we scaled and combined the features identified as most impactful for gene segmentation. Specifically, we increased the input sequence length to 250 Kb, utilized data from 39 mammalian species, and included all gene types in the training set. For the Transformer-based architecture, we employed a larger instance of `GENA-LM` with an increased number of parameters (`GENA large`), while for the SSM we used the `Caduceus PS` variant, which consistently demonstrated performance superior to `Caduceus PH` in our benchmarks. We deliberately conducted most experiments on a small dataset with downscaled models to conserve computational resources while reporting detailed usage statistics (see Appendix J).

At test time we applied both the splice-site filtering and RC augmentation strategies. We refer to the resulting models as `GENATATOR`s, a DNA language model-based family of gene annotators.

Both `GENA large` and `Caduceus PS` show significant performance improvements after scaling (Figure 1). Interestingly, the performance gain was more pronounced for `GENA large`, resulting in a higher overall segmentation accuracy compared to `Caduceus PS`. This contrast in model ranking after scaling may be attributed to two factors. First, the increase in model size was feasible only for `GENA-LM` because a larger pre-trained instance was available, whereas no larger variant of `Caduceus` currently exists. Second, the Recurrent Memory Transformer architecture employed in `GENA-LM` provides a superior ability to handle long input sequences in comparison with SSMs (Rodkin et al., 2025).

Among gene types, the previously observed specificity of each model remained consistent after scaling. `GENA large` achieved superior performance in the segmentation of lncRNAs, while `Caduceus PS` continued to outperform in the detection of protein-coding gene structure and in the accurate annotation of CDS (Figure 1 and Appendix G Table A14).

### 4.4 BENCHMARKING GENATATOR AGAINST OTHER GENE-ANNOTATION TOOLS

We evaluated the performance of the `GENATATOR` models in comparison with several state-of-the-art gene annotation tools, including the HMM-based `AUGUSTUS` (Stanke et al., 2004), the CNN+HMM model `Tiberius` (Gabriel et al., 2024), the DNA LM-based `SegmentNT` (with variants trained on human-only and multispecies data) (de Almeida et al., 2025), and transformer-based models pretrained on gene expression, namely `SegmentEnformer` and `SegmentBorzoi` (de Almeida et al., 2025). We also included the recently developed `AlphaGenome` in the comparison (Avsec et al., 2025).

We first compared models using the conventional PR-AUC metric (Appendix G Table A12). According to this evaluation, `GENATATOR`s slightly outperform `SegmentNT`, `SegmentBorzoi`, and `SegmentEnformer`, with an improvement of about 10% between the best-performing `GENATATOR` and the best-performing `SegmentNT`.

We then assessed performance using gene level metrics described above, reporting results as the total number of correctly segmented genes (Figure 1, detailed metrics and model usage in Appendix K). Under this scoring scheme, `GENATATOR`s identify substantially more genes, with more than a threefold difference compared to previously developed alternatives. Visual inspection of predicted gene structures reveals that `SegmentNT` frequently extends exon boundaries by several nucleotides, which in the case of mRNA leads to reading-frame shifts and translates to biologically invalid truncated peptides. This observation underscores the importance of gene level evaluation metrics for capturing biologically meaningful segmentation accuracy.

We attribute the improved performance of `GENATATOR`s to a combination of training optimizations, including the use of multispecies data, extended input context lengths, and data augmentation strategies. As shown in Table 1, a basic training configuration with human-only data, a 4,096 bp input length, and no augmentations, or isolated modifications of this setup, produces results that are comparable to or worse than those achieved by `SegmentNT`.

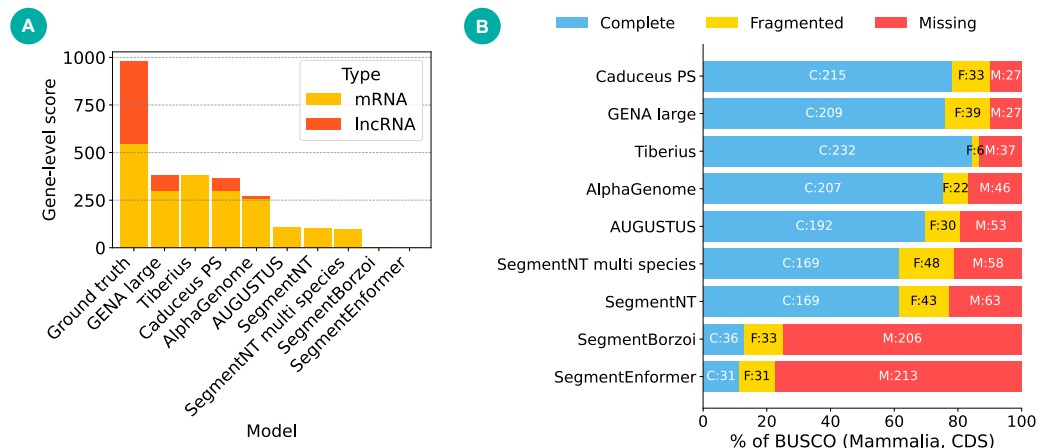

Figure 1: GENATATORs are top-ranked in gene segmentation benchmarks. A. Gene-level metrics. B. BUSCO metrics

GENATATORs also outperform AUGUSTUS in the total number of correctly segmented genes and perform on par with the current state-of-the-art model Tiberius. Specifically, the GENA-based GENATATOR marginally outperforms Tiberius, while the Caduceus-based variant performs slightly below it.

Stratifying performance by transcript type reveals that Tiberius outperforms both GENATATORs in the number of correctly segmented protein-coding regions of genes, which stem from its superior performance in CDS classification. However Tiberius completely fails to identify lncRNA genes and UTRs within mRNA genes, resulting in slightly lower total number of correctly segmented genes.

The common metric for assessing the completeness of genome annotation is BUSCO (Manni et al., 2021). To compute BUSCO, the predicted exon-intron structure of a gene is used to generate an amino acid sequence, which is then compared to a set of proteins that are specific to a particular taxonomy group. The results of BUSCO are presented as a number of proteins that were identified from a selected dataset. These proteins are divided into two categories: Complete and Fragmented, where fragmented proteins have some segments missing.

Using the mammalia-specific BUSCO dataset, GENATATORs identified 246 orthologs, outperforming all other models. Tiberius detected 238 orthologs, but with a higher number of complete genes (232 for Tiberius vs. 210 for GENATATOR). Similar trends were observed using the primates BUSCO dataset.

Other models, including SegmentNT, SegmentBorzoi, and SegmentEnformer, showed substantially lower BUSCO recovery rates, consistent with their lower gene level segmentation performance. These results further reinforce the conclusion that conventional classification metrics such as PR-AUC are poor proxies for evaluating biological utility of the models.

We next investigated whether segmentation errors made by different tools are shared or model-specific. Shared errors would suggest the presence of genes with structural features that are out-of-distribution relative to the training data, while model-specific errors would indicate that each tool fails on a unique subset of genes. To explore this, we analyzed the overlap of correctly segmented genes among the three top-performing models: the two GENATATOR variants and Tiberius. As shown in Appendix G Figure A2, there is a substantial intersection of correctly segmented genes across all models, supporting the hypothesis that certain genes present a challenge to all tools. At the same time, each model also segments a distinct subset of genes not correctly annotated by the others. In comparisons between GENATATORs and Tiberius, the unique gene set recovered by GENATATORs is largely composed of lncRNAs, which Tiberius is not designed to annotate. These findings suggest that model ensembling is currently the most effective strategy for maximizing gene annotation coverage across both coding and non-coding transcripts.

Overall, our results position GENATATORs as state-of-the-art models for gene annotation, with particular strength in the detection of non-coding genes and UTRs.

## 4.5 GENATATORs GENERALIZE ACROSS UNSEEN SPECIES AT LARGE EVOLUTIONARY DISTANCES

Table 2: Gene-level performance of different models on evolutionarily distant species.

| Species | Chromosome | Gene type | Class | Caduceus PS (%) | GENA large (%) | Tiberius (%) | AUGUSTUS (%) |
|---------|-----------|-----------|-------|-----------------|----------------|--------------|--------------|
| | | mRNA | EXON | 26.56 | **30.59** | 0.10 | 7.59 |
| *A. thaliana* | NC_003075.7 | | CDS | 14.80 | 8.43 | 14.06 | **55.33** |
| | | lncRNA | EXON | 41.13 | **60.04** | 0.00 | 0.39 |
| | | all RNA | EXON | 28.16 | **33.81** | 0.09 | 6.80 |
| | | mRNA | EXON | **96.21** | 90.99 | 0.00 | 0.00 |
| *S. cerevisiae* | NC_001136.10 | | CDS | **94.13** | 89.95 | 0.00 | 46.74 |
| | | lncRNA | EXON | NA | NA | NA | NA |
| | | all RNA | EXON | **96.21** | 90.99 | 0.00 | 0.00 |

A key application of *ab initio* gene predictors is the annotation of genomes from previously unannotated species. To evaluate the cross-species generalization of our models, we first evaluated performance using gene-level metric on two evolutionarily remote species representing different kingdoms of life: the flowering plant *Arabidopsis thaliana* (GCF_000001735.4) and the budding yeast *Saccharomyces cerevisiae* (GCF_000146045.2) (Table 2). At the nucleotide level, there is effectively no sequence homology between their genes and those of mammalian species included in the training dataset, and thus the models had never encountered any comparable sequences during training. Despite this extreme divergence, the models retained reasonable accuracy. For *A. thaliana*, GENA large correctly reconstructed approximately one-third of all exons and over 60% of lncRNA exons, far surpassing AUGUSTUS and Tiberius. For *S. cerevisiae*, whose compact genome lacks spliceosomal introns, Caduceus PS achieved 96% exon recall and 94% CDS recall, substantially outperforming both baselines. NA entries in the lncRNA row of Table 2 indicate the absence of annotated lncRNAs in the reference genome. Same results were obtained when we excluded all genes with detectable protein-level similarity to mammals, to ensure that model's can not find homology even after internally translating DNA to amino acid code. Under this stringent setting, GENATATORs reconstructed more than twice as many genes as AUGUSTUS, despite the latter being run with a species-specific profile (Appendix L). Thus, although not tuned for plants or fungi, the models were able to produce useful first-pass annotations in such genomes, providing strong evidence that their capabilities extend beyond mere memorization of homologous patterns.

In addition to this extreme test, we benchmarked the models across a spectrum of animal species, ranging from primates closely related to humans to distant lineages such as insects (Appendix M). The relative ranking of methods remained consistent across these taxa: GENATATORs and Tiberius consistently outperformed other baselines, with DNA LMs showing superior generalization on more distant organisms. For protein-coding genes, segmentation accuracy gradually decreased with evolutionary distance, whereas for lncRNAs, performance remained in the range of 10-30% across all species, with GENA-based architectures consistently outperforming Caduceus-based ones.

## CONCLUSIONS

In this work, we comprehensively evaluated the utility of DNA LMs for the gene segmentation task. We show, both theoretically and empirically, that interval level metrics better reflect biological relevance than conventional token level classifiers and introduce dedicated benchmark to score gene segmentation models.

We demonstrated that embeddings from pretrained DNA LMs do not contain sufficient information for accurate gene segmentation. However, by identifying optimal training regimes, datasets, augmentations, and output filters, we enabled efficient fine-tuning and inference of gene structure. We further showed that scaling DNA LMs under these conditions substantially improves performance leading to state-of-the-art results.

We found that sensitivity to different functional gene elements—such as CDS and UTRs—varies across DNA LM architectures. Nonetheless, all evaluated DNA LMs were capable of detecting lncRNA genes, which remain inaccessible to current state-of-the-art tools such as Tiberius.

Furthermore, GENATATORs, our fine-tuned DNA LM-based models, generalize effectively to unseen species across large evolutionary distances. These results highlight the potential of DNA LMs to serve as powerful tools for *de novo* genome annotation in a wide range of biological and evolutionary studies. We discuss limitations of this work in Appendix N.

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

## APPENDIX A. GENE STRUCTURE AND SEGMENTATION PROBLEM.

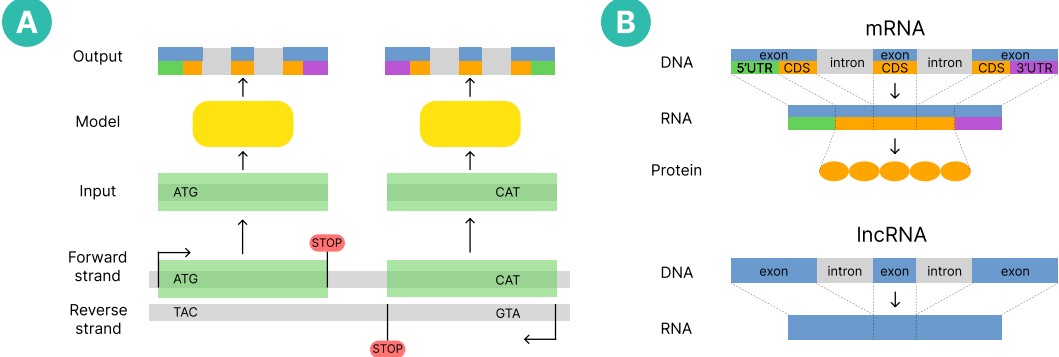

Figure A1: Gene structure and segmentation problem. Panel A shows transcript types in the dataset, where the model predicts all five classes but only intron and exon labels are relevant for lncRNAs, while all five are meaningful for mRNAs. Panel B illustrates that the model always receives DNA sequence from the forward strand (light green box) during training, yet these sequences may correspond to genes located on either strand.

APPENDIX B. DIFFERENCES BETWEEN OUR BENCHMARK AND OTHERS.

This appendix compares our benchmark with BEND (Marin et al., 2023) and with GUE introduced alongside DNABERT-2 (Zhou et al., 2023), focusing on input length coverage, task granularity, and the biological meaning of reported metrics. Table A1 summarizes the design choices in each suite, and Table A2 reports human training-set lengths that illustrate coverage differences.

Table A1: Design comparison of benchmarks.

| Benchmark | Input scope | Typical length | Granularity | Evaluation scope | Metrics |
|---|---|---|---|---|---|
| GUE (Zhou et al., 2023) | short sequences | 70–1000 bp; splice sites 400 bp; GUE+ 5–10 kb | sequence-level | local classification tasks | task-specific (MCC / F1) |
| BEND (Marin et al., 2023) | gene snippets | up to 13 kb | nucleotide-level | nucleotide classification of gene-structure labels; no full-gene segmentation; no UTR / lncRNA | MCC only |
| Ours | full genes via tiling | train 4 096 or 32 k or 250 k nt; full gene length evaluation | nucleotide-level | end-to-end segmentation with full gene reconstruction; with UTR and lncRNA | interval-, gene-level |

A key difference is length coverage and how it affects evaluation. As summarized in Table A2, our training data span substantially longer transcripts than BEND, preserving the long tail of gene lengths; in fact, 17,737 human transcripts in our set exceed 13,000 nt, whereas BEND truncates at this length. In addition, sequence-level suites such as GUE emphasize short-range classification and report scores that do not capture boundary accuracy, while BEND, although nucleotide-level, uses metrics that are not biologically rigorous for full gene structures and does not assess UTRs or lncRNA genes. By contrast, our evaluation targets complete gene structures with interval- and gene-level metrics; a detailed analysis of metric sensitivity appears in Appendix C.

Table A2: Statistics of training datasets for BEND and our benchmark (human).

| Dataset | # Transcripts | Mean length (nt) | Median (nt) | 95th perc. (nt) | Max (nt) |
|---|---|---|---|---|---|
| BEND (human) | 4,783 | 7,474 | 7,355 | 12,414 | 13,000 |
| Our (human) | 33,367 | 37,366 | 14,651 | 176,543 | 250,000 |

**Benchmarking on BEND**  For comparability we also report results on BEND. Unlike the probing setup in the original BEND paper that assesses the quality of the embeddings in different pretrained models, we fine-tuned our models until convergence using the official train, validation, and test splits. This decision was deliberate: BEND compared all models against AUGUSTUS, which is a trained HMM genome annotation tool (it saw all human genes in the BEND benchmark during training). To ensure fairness we therefore also trained our models. Because sequences in BEND are short, all of our models can handle the full length of each sample, so no chunking was applied at either training or validation. The reported metric is MCC, as specified in the BEND paper.

**Comparison of pur benchmark with G3PO and Tiberius approaches**  The G3PO benchmark (Scalzitti et al., 2020) is constructed from 1,793 UniProt proteins grouped into twenty orthologous families selected to represent complex protein-coding genes across 147 species. For each protein, the corresponding genomic locus and exon map are retrieved from Ensembl, and evaluation is carried out at nucleotide, exon, and protein levels against a single reference protein per gene. Consequently, G3PO covers only protein-coding genes, excludes lncRNA, and does not assess complete gene structure across multiple transcript isoforms.

Table A3: BEND gene-finding results (MCC) with fine-tuned models using official splits and full-sequence inference.

| Model | MCC |
|---|---|
| Caduceus PS | **0.83** |
| AUGUSTUS | 0.80 |
| Caduceus PH | 0.72 |
| GENA base | 0.65 |

Tiberius (Gabriel et al., 2024) is trained on mammalian protein-coding genes and uses convolutional and recurrent layers combined with a differentiable HMM. To obtain unambiguous labels, only the transcript with the longest coding sequence is retained for each gene, and evaluation is performed against this single coding isoform. As a result, exon- and gene-level metrics for Tiberius are computed relative to one reference isoform rather than across the full isoform set.

In contrast, our benchmark evaluates complete exon structures for all supported transcript types, including UTR exons, coding exons, and exons of lncRNAs. A prediction is counted as correct only when the full set of predicted exons matches the exon set of at least one annotated isoform, which allows transcripts containing both coding and non-coding segments to be evaluated faithfully. Together with CDS-based metrics comparable to those used in Scalzitti et al. (2020), our interval- and gene-level metrics provide a more biologically aligned assessment of complete gene reconstruction for both coding and non-coding genes.

APPENDIX C. PR-AUC SENSITIVITY AND SUPPORTING EVIDENCE.

## C.1 THEORETICAL EVIDENCE

Per nucleotide metrics such as precision, recall, f1 and PR-AUC treat each base independently, which can hide small local mistakes that have large biological impact. We provide theoretical evidence of this discrepancy between nucleotide and interval level metrics using a binary setup with two mutually exclusive classes, exon coded as 1 and intron coded as 0. For a single gene containing $p$ positive exon bases and $n$ negative intron bases with positive scores $s_i$, PR-AUC equals Average Precision and can be written using the ranks of positives in the list sorted by $s_i$ in descending order

$$\text{PR-AUC} = \text{AP} = \frac{1}{p} \sum_{k \in R_+} \text{Pr}(k), \qquad \text{Pr}(k) = \frac{\#\text{positives in top } k}{k}, \tag{3}$$

where $R_+$ is the set of positions in the sorted list that are occupied by positives. This depends only on the ordering of scores, so any monotone transformation that preserves order keeps PR-AUC unchanged.

We now carry one simple example through the derivation so that each step is explicit. Consider a short gene with a single exon block followed by an intron block. The targets and baseline scores are

$$y = \begin{bmatrix} 1, 1, 1, 1, 0, 0, 0, 0 \end{bmatrix} \quad \text{and} \quad s = \begin{bmatrix} 0.99, 0.95, 0.92, 0.91, 0.40, 0.35, 0.31, 0.20 \end{bmatrix}.$$

This is a good prediction because exons receive higher scores than introns. The scores are already in descending order, so the cumulative number of exons in the top $k$ positions is

$$T(1) = 1,\ T(2) = 2,\ T(3) = 3,\ T(4) = 4,\ T(5) = 4,\ T(6) = 4,\ T(7) = 4,\ T(8) = 4,$$

and the corresponding precision values are

$$\text{Pr}(1) = \frac{1}{1},\ \text{Pr}(2) = \frac{2}{2},\ \text{Pr}(3) = \frac{3}{3},\ \text{Pr}(4) = \frac{4}{4},\ \text{Pr}(5) = \frac{4}{5},\ \text{Pr}(6) = \frac{4}{6},\ \text{Pr}(7) = \frac{4}{7},\ \text{Pr}(8) = \frac{4}{8}.$$

Average Precision averages these precision values only at the positive positions $k \in \{1, 2, 3, 4\}$, hence

$$\text{AP} = \frac{1}{4}\left(1 + \frac{2}{2} + \frac{3}{3} + \frac{4}{4}\right) = 1. \tag{4}$$

If we apply a monotone change to all scores, for example $s \mapsto s^2$ or $s \mapsto s + 5$, the order does not change and equation 4 remains the same, which illustrates the order invariance of PR-AUC in equation 3.

We now introduce a boundary error at the exon edge before sorting and we make the modification explicit. Keep the targets $y$ fixed and lower the scores of the last two exon bases so that they fall below all intron scores. Define the modified score vector

$$\tilde{s} = \begin{bmatrix} 0.99, 0.95, \underline{0.19}, \underline{0.18}, 0.40, 0.35, 0.31, 0.20 \end{bmatrix},$$

where the underlined entries mark the two exon bases affected by the boundary error. This change is applied before sorting by score. After sorting $\tilde{s}$ in descending order, the new score order is

$$\tilde{s}_{\text{sorted}} = \begin{bmatrix} 0.99, 0.95, 0.40, 0.35, 0.31, 0.20, 0.19, 0.18 \end{bmatrix},$$

and the corresponding sorted labels become

$$y'_{\text{sorted}} = \begin{bmatrix} 1, 1, 0, 0, 0, 0, 1, 1 \end{bmatrix}.$$

Thus the two undemoted exons stay at ranks 1 and 2, the four introns occupy ranks 3 through 6, and the two demoted exons move to ranks 7 and 8. The cumulative positives for the modified order are

$$T'(1) = 1,\ T'(2) = 2,\ T'(3) = 2,\ T'(4) = 2,\ T'(5) = 2,\ T'(6) = 2,\ T'(7) = 3,\ T'(8) = 4,$$

and the Average Precision after the error averages the precision values at the positive ranks $1, 2, 7, 8$

$$\text{AP}' = \frac{1}{4}\left(\frac{1}{1} + \frac{2}{2} + \frac{3}{7} + \frac{4}{8}\right) = \frac{1}{4}\left(1 + 1 + \frac{3}{7} + \frac{1}{2}\right) = \frac{41}{56} \approx 0.7321.$$

We now connect this explicit computation with the general formula. In the general case with $p$ exon nucleotides and $n$ intron nucleotides, if $\delta$ exon bases near the boundary are lowered below all intron

scores before sorting, the sorted list contains $p - \delta$ exons first, then $n$ introns, then the $\delta$ demoted exons. The $r$th demoted exon occupies rank

$$k_r = n + (p - \delta) + r \quad \text{for} \quad r = 1, \ldots, \delta,$$

because the top contains $p - \delta$ undemoted exons and $n$ introns before the first demoted exon appears. At rank $k_r$ the prefix contains $(p - \delta) + r$ exons, so its precision equals

$$\text{Pr}(k_r) = \frac{p - \delta + r}{n + p - \delta + r}.$$

All remaining $p - \delta$ exons at ranks 1 through $p - \delta$ have precision 1. Plugging these two groups into equation 3 gives the exact PR-AUC after the boundary error

$$\text{PR-AUC}' = \frac{1}{p}\Big[(p - \delta) \cdot 1 + \sum_{r=1}^{\delta} \frac{p - \delta + r}{n + p - \delta + r}\Big]. \tag{5}$$

For the example with $p = 4$, $n = 4$ and $\delta = 2$ this yields

$$\text{PR-AUC}' = \frac{1}{4}\Big[2 \cdot 1 + \frac{3}{7} + \frac{4}{8}\Big] = \frac{41}{56},$$

which is exactly the value computed from the sorted example above.

The corresponding loss is

$$\Delta\text{PR-AUC} = 1 - \text{PR-AUC}' = \frac{1}{p}\sum_{r=1}^{\delta}\Big(1 - \frac{p - \delta + r}{n + p - \delta + r}\Big)$$

$$= \frac{1}{p}\sum_{r=1}^{\delta} \frac{n}{n + p - \delta + r}$$

$$\leq \frac{1}{p}\sum_{r=1}^{\delta} \frac{n}{n + 1} = \frac{\delta\, n}{p\,(n + 1)} \leq \frac{\delta}{p}. \tag{6}$$

The last two inequalities hold because each denominator satisfies $n + p - \delta + r \geq n + 1$, hence each summand is at most $n/(n + 1) < 1$, so the sum of $\delta$ such terms is at most $\delta\, n/(n + 1) < \delta$, and dividing by $p$ yields the stated bound $\Delta\text{PR-AUC} \leq \delta/p$.

Under the same error the interval and gene views behave differently. If the gene has $m$ true exon intervals and the boundary of one interval moves by one base, that interval no longer matches exactly. True positives drop from $m$ to $m - 1$ and at least one false positive and one false negative appear. Substituting into Eq. equation 1 yields

$$\text{F1}_{\text{interval}}^{\text{exon}} = \frac{2(m - 1)}{2(m - 1) + 2} = 1 - \frac{1}{m}. \tag{7}$$

Define the interval drop as the difference between the perfect and the post error score. With one boundary shift that breaks exactly one interval and introduces exactly one false positive and one false negative, the drop is

$$\Delta\text{F1}_{\text{interval}}^{\text{exon}} = 1 - \Big(1 - \frac{1}{m}\Big) = \frac{1}{m}, \tag{8}$$

and it can be larger if the prediction creates additional spurious or missed intervals.

At gene level the same single boundary shift breaks the exact match for all isoforms, so the gene contributes 1 before the error and 0 after

$$\Delta\text{Score}_{\text{gene}} = 1. \tag{9}$$

Given equation 6, equation 8 and equation 9, the sensitivity fractions for the same local error satisfy

$$\frac{\Delta\text{PR-AUC}}{\Delta\text{Score}_{\text{gene}}} \leq \frac{\delta}{p} \quad \text{and} \quad \frac{\Delta\text{PR-AUC}}{\Delta\text{F1}_{\text{interval}}^{\text{exon}}} \leq \frac{\delta}{p}\, m. \tag{10}$$

With $m$ fixed and $p$ large the right hand sides are small. Therefore, given the same boundary mistake, PR-AUC changes by at most $\delta$ over $p$ and becomes negligible on long exons, while the interval score and the gene score incur fixed drops per affected interval and per affected gene.

## C.2 EMPIRICAL EVIDENCE

We complement the theory with experiments scoring models with PR-AUC and interval level metrics (Table A4). These results show that model ranking depends on the metrics used..

Table A4: Why gene level metrics matter, comparison of mean PR-AUC and fully reconstructed genes

| model | PR-AUC mean | gene level all |
|---|---|---|
| Caduceus PH 32 kb | 0.656 | 120 |
| Caduceus PS 32 kb | 0.668 | 130 |
| GENA-LM 250 kb | 0.635 | 383 |

Both Caduceus variants exceed GENA-LM by PR-AUC mean, yet they reconstruct about three times fewer genes, since 130 versus 383. Across all models the spread in mean PR-AUC is about $0.16$, for example Caduceus PS $0.680$, SegmentNT $0.611$, SegmentEnformer $0.520$, while the difference in fully reconstructed genes ranges from 0 to 383. With these numbers in mind, optimizing only PR-AUC during early experiments can reward architectures that seem promising while failing to assemble biologically valid transcripts, which slows progress.

We further trained models on a human gene set with the same labels but one label per BPE token and varied input length from $4k$ BPE tokens which is approximately 32k nucleotides to 32k tokens which is approximately 250k nucleotides.

Table A5: Effect of input length and output granularity on PR-AUC mean and gene level all for GENA large

| setting | input length nt | PR-AUC mean | gene level all |
|---|---|---|---|
| 4 k | $\approx 32k$ | 0.628 | 44 |
| 16 k | $\approx 128k$ | 0.642 | 66 |
| 32 k (BPE) | $\approx 250k$ | 0.648 | 106 |
| 32 k (nucleotide, human) | $\approx 250k$ | 0.672 | 208 |

Mean PR-AUC differs by about $0.020$ between the 4k and 32k BPE models, yet the gene level score rises from 44 to 106 which is a factor of about 2.5. Switching from BPE outputs to nucleotide outputs by stacking a UNET on top of the trained model changes PR-AUC from $0.648$ to $0.672$, while the number of fully reconstructed genes increases by 102 which is a factor of about 2. With the arguments provided in Section C.1 and these empirical trends, we get that context length and boundary precision both matter for transcript assembly and that interval and gene level evaluation is needed when developing annotation models.

## APPENDIX D. DATASET PREPARATION, MODEL TRAINING AND ARCHITECTURE DETAILS

The dataset was constructed using the human genome assembly GCF_009914755.1. Chromosomes 8, 20 and 21 were designated as the validation set, but only chromosome 20 was used to compute final metrics for computational efficiency. We did not use a separate test set. The dataset contains all mRNA and lncRNA genes, and all sequences were exclusively from the forward strand. The dataset was filtered via selecting one representative transcript per gene, choosing the longest transcript available. Only transcripts with a length of up to 250 Kb were included.

Below we provide details of modifications in dataset, training regime or architecture for specific models:

1. For the mRNA-only dataset, we selected samples corresponding exclusively to protein-coding genes from the original dataset.

2. For the multispecies dataset, we processed data for 39 species (38 plus human) using the same strategy as for human samples. The list of species is provided in Table A6. It's important to note that only the human genome is fully assembled, therefore samples from other species containing 'N' characters (indicating unknown sequences) were excluded.

3. All models were trained using flash attention support (if supported by the model) to improve computational efficiency.

4. For training BPE-based GENA models at nucleotide-level resolution, embeddings derived from the token-level models were employed, omitting memory, CLS, and SEP tokens. The primary distinction between handling embeddings from GENA-LM versus other models arises from GENA-LM's use of BPE tokens, necessitating additional steps before U-Net usage, whereas models like Caduceus and Evo2 already operate directly at nucleotide resolution. Specifically, for GENA-LM, token embeddings were upsampled, meaning each embedding was replicated according to how many nucleotides it covered. Subsequently, nucleotide-specific embeddings (one per nucleotide type, totally four different learnable embeddings) were concatenated to these upsampled token embeddings. For computational efficiency, those embeddings were segmented into non-overlapping chunks of 8192 base pairs (along sequence length axis), which were individually fed into the U-Net model. In contrast, for models that can directly utilize nucleotide resolution, we simply included an additional fully connected layer to convert embeddings into class probability vectors.

5. A learning rate of $5 \times 10^{-5}$ and weight decay of $1 \times 10^{-4}$ with AdamW optimizer was discovered to be the optimal trade-off between prediction accuracy (particularly for splice site boundary detection) and convergence speed, as lower values adversely impacted prediction quality.

6. Training of each model was performed on 8 Nvidia GPUs (either A100 or H100), except for Evo2, which specifically required Nvidia H100 GPUs due to compatibility constraints (GPU compatibility $> 8.9$). All models were trained until convergence was observed using an exon-level validation metric. Typically, training with frozen embeddings required approximately half a day, while low-scale finetuning took about two days, with slight variations depending on the specific model. It took us one week to train the final models presented in our benchmark section.

7. In training and internal validation we do not always take nucleotides from the beginning of a gene. Instead, we choose a random starting position and extract at most $N$ nucleotides to the right, where $N$ is the model's context length (4096, 32k, or 250k as reported in the main text). We also ensure that the selected subsequence is at least 512 nucleotides long, so that the model always receives enough context. Each gene contributes a single subsequence of this form, with no splitting. Metrics computed in this setup, such as AUC and interval level scores, are used only to select the best checkpoint for later evaluation.

8. For the final validation reported in the paper we evaluate complete genes. Here, sequences are divided into non-overlapping chunks of the same length that the model was trained on. Predictions are made for each chunk, then concatenated to recover the full gene, and metrics are calculated on the full-gene predictions. This guarantees consistency with training while still allowing evaluation of arbitrarily long genes.

Table A6: List of genomic assemblies used to create the multispecies training dataset. List of genomic assemblies used to create the multispecies training dataset. Assembly names correspond to the annotation and genome names. The annotation files have been received by the NCBI Eukaryotic Genome Annotation Pipeline.

| Assembly | Species |
|---|---|
| GCF_000952055.2 | Aotus nancymaae |
| GCF_002263795.3 | Bos taurus |
| GCF_000767855.1 | Camelus bactrianus |
| GCF_000002285.3 | Canis lupus familiaris |
| GCF_000151735.1 | Cavia porcellus |
| GCF_001604975.1 | Cebus imitator |
| GCF_000283155.1 | Ceratotherium simum simum |
| GCF_000276665.1 | Chinchilla lanigera |
| GCF_000260355.1 | Condylura cristata |
| GCF_002940915.1 | Desmodus rotundus |
| GCF_000151885.1 | Dipodomys ordii |
| GCF_002288905.1 | Enhydra lutris kenyon |
| GCF_000308155.1 | Eptesicus fuscus |
| GCF_000002305.2 | Equus caballus |
| GCF_018350175.1 | Felis catus |
| GCF_000247695.1 | Heterocephalus glaber |
| GCF_009914755.1 | Homo sapiens |
| GCF_000236235.1 | Ictidomys tridecemlineatus |
| GCF_000280705.1 | Jaculus jaculus |
| GCF_000001905.1 | Loxodonta africana |
| GCF_001458135.1 | Marmota marmota |
| GCF_000165445.2 | Microcebus murinus |
| GCF_000317375.1 | Microtus ochrogaster |
| GCF_000001635.26 | Mus musculus |
| GCF_900095145.1 | Mus pahari |
| GCF_002201575.1 | Neomonachus schauinslandi |
| GCF_000292845.1 | Ochotona princeps |
| GCF_000260255.1 | Octodon degus |
| GCF_000321225.1 | Odobenus rosmarus divergens |
| GCF_009806435.1 | Oryctolagus cuniculus |
| GCF_000181295.1 | Otolemur garnettii |
| GCF_016772045.2 | Ovis aries |
| GCF_000956105.1 | Propithecus coquereli |
| GCF_003327715.1 | Puma concolor |
| GCF_036323735.1 | Rattus norvegicus |
| GCF_000235385.1 | Saimiri boliviensis boliviensis |
| GCF_000181275.1 | Sorex araneus |
| GCF_000003025.6 | Sus scrofa |
| GCF_000243295.1 | Trichechus manatus latirostris |

APPENDIX E. COMPARISON OF MODELS FOR *de novo* GENE ANNOTATION

Table A7: Comparison of Classical, State-of-the-Art, and Emerging Models for *de novo* Gene Annotation

| Model | Architecture (details) | N params, M | Input, Kb | Tokenization | Released |
|---|---|---|---|---|---|
| AUGUSTUS | HMM | N/A | N/A | 1-bp | (Stanke et al., 2004) |
| Tiberius | CNN+HMM | 8 | 10 | 1-hot | (Gabriel et al., 2024) |
| SegmentNT | Transformer (RoPE) + UNET | 500 | 50 | 6-mer | (de Almeida et al., 2025) |
| SegmentEnformer/Borzoi | Transformer + UNET | 200 | 50 | 1-bp | (de Almeida et al., 2025) |
| AlphaGenome | CNN + Transformer | 450 | 1000 | 1-bp | (Avsec et al., 2025) |
| GENATATOR (GENA large) | Transformer (RMT) + UNET | 360 | 250 | BPE | this work |
| GENATATOR (GENA base) | Transformer (RMT) + UNET | 120 | 32 | BPE | this work |
| GENATATOR (Caduceus PH) | SSM | 15 | 250 | nucleotide | this work |
| GENATATOR (Caduceus PS) | SSM (+RC equivalent) | 15 | 250 | nucleotide | this work |
| GENATATOR (Evo) | SSM (S3 layers) | 1000 | 32 | nucleotide | this work (probing only) |
| SegmentBorzoi | CNN + UNET | 323 | 196 | nucleotide | this work |
| SegmentEnformer | Transformer + UNET | 379 | 196 | nucleotide | this work |

## APPENDIX F. TRAINING ON EMBEDDINGS.

Table A8: Gene-level metric after training on frozen embeddings of different DNA LM models.

| Chunk length ($N_l$), bp | 4096 | | | | 32000 | | | |
|---|---|---|---|---|---|---|---|---|
| Performance (gene-lvl metrics) | mRNA | | lncRNA | all RNA | mRNA | | lncRNA | all RNA |
| | exon | CDS | exon | exon | exon | CDS | exon | exon |
| GENA base | 4 | 0 | 1 | 5 | 7 | 0 | 2 | 9 |
| Caduceus PH | 0 | 0 | 0 | 0 | 0 | 0 | 0 | 0 |
| Caduceus PS | 0 | 0 | 0 | 0 | 0 | 0 | 0 | 0 |
| Evo2 | 0 | 0 | 0 | 0 | 0 | 0 | 0 | 0 |

## APPENDIX G. MODELS SCORING AND BENCHMARKING.

Table A9: Interval level metrics related to Table A8 (embedding training). Data shown for exon and CDS class.

| Model/train setup | | | 4096 | | | 32000 | | |
|---|---|---|---|---|---|---|---|---|
| | | | precision | recall | f1 | precision | recall | f1 |
| GENA base | mRNA | exon | 0.0077 | 0.1124 | 0.0145 | 0.0023 | 0.0096 | 0.0037 |
| | | CDS | 0.0197 | 0.0655 | 0.0303 | 0.0013 | 0.0029 | 0.0018 |
| | lncRNA | exon | 0.0032 | 0.0440 | 0.0060 | 0.0011 | 0.0059 | 0.0019 |
| | all RNA | exon | 0.0068 | 0.0969 | 0.0127 | 0.0020 | 0.0088 | 0.0032 |
| Caduceus PH | mRNA | exon | 0.0000 | 0.0000 | 0.0000 | 0.0000 | 0.0000 | 0.0000 |
| | | CDS | 0.0000 | 0.0000 | 0.0000 | 0.0000 | 0.0000 | 0.0000 |
| | lncRNA | exon | 0.0000 | 0.0000 | 0.0000 | 0.0000 | 0.0000 | 0.0000 |
| | all RNA | exon | 0.0000 | 0.0000 | 0.0000 | 0.0000 | 0.0000 | 0.0000 |
| Caduceus PS | mRNA | exon | 0.0000 | 0.0000 | 0.0000 | 0.0000 | 0.0000 | 0.0000 |
| | | CDS | 0.0000 | 0.0000 | 0.0000 | 0.0000 | 0.0000 | 0.0000 |
| | lncRNA | exon | 0.0000 | 0.0000 | 0.0000 | 0.0000 | 0.0000 | 0.0000 |
| | all RNA | exon | 0.0000 | 0.0000 | 0.0000 | 0.0000 | 0.0000 | 0.0000 |
| Evo2 | mRNA | exon | 0.0000 | 0.0000 | 0.0000 | 0.0000 | 0.0000 | 0.0000 |
| | | CDS | 0.0000 | 0.0000 | 0.0000 | 0.0000 | 0.0000 | 0.0000 |
| | lncRNA | exon | 0.0000 | 0.0000 | 0.0000 | 0.0000 | 0.0000 | 0.0000 |
| | all RNA | exon | 0.0000 | 0.0000 | 0.0000 | 0.0000 | 0.0000 | 0.0000 |

Table A10: Comparison of GENA base (Table 1) and GENA large in the baseline setup.

| Model | Category | Gene-level |
|---|---|---|
| GENA base | EXON mRNA + lncRNA | 46 |
| | EXON mRNA | 31 |
| | EXON lncRNA | 15 |
| | CDS mRNA | 1 |
| GENA large | EXON mRNA + lncRNA | 61 |
| | EXON mRNA | 42 |
| | EXON lncRNA | 19 |
| | CDS mRNA | 5 |

Table A11: Interval level performace metrics for small-scale finetunning experiments with dataset and model modifications (absolute scores obtained for each setup), related to Table 1. Data shown for exon and CDS classes.

| Sequnce length | | | 4096 | | | 32000 | | | mRNA | | | 4096 | | |
|---|---|---|---|---|---|---|---|---|---|---|---|---|---|---|
| Gene type | | | all | | | all | | | mRNA | | | all | | |
| Species | | | human | | | human | | | human | | | 39 mammals | | |
| Test-time augmentation | | | no | | | no | | | no | | | no | | |
| Model/dataset | | | precision | recall | f1 | precision | recall | f1 | precision | recall | f1 | precision | recall | f1 |
| GENA base | mRNA | exon | 0.0452 | 0.4110 | 0.0815 | 0.1746 | 0.5906 | 0.2695 | 0.0340 | 0.3890 | 0.0625 | 0.0630 | 0.5321 | 0.1127 |
| | | CDS | 0.0489 | 0.4452 | 0.0881 | 0.1976 | 0.6570 | 0.3038 | 0.0394 | 0.4448 | 0.0723 | 0.0698 | 0.5914 | 0.1249 |
| | lncRNA | exon | 0.0327 | 0.2945 | 0.0588 | 0.1016 | 0.3636 | 0.1588 | 0.0166 | 0.1986 | 0.0307 | 0.0395 | 0.3294 | 0.0705 |
| | all RNA | exon | 0.0821 | 0.1875 | 0.1142 | 0.2876 | 0.5178 | 0.3698 | 0.0538 | 0.3616 | 0.0936 | 0.0651 | 0.4451 | 0.1136 |
| Caduceus PH | mRNA | exon | 0.1168 | 0.5595 | 0.1932 | 0.1912 | 0.6251 | 0.2928 | 0.2060 | 0.5602 | 0.3012 | 0.1428 | 0.6711 | 0.2355 |
| | | CDS | 0.1524 | 0.6642 | 0.2479 | 0.2577 | 0.7263 | 0.3804 | 0.2719 | 0.6884 | 0.3899 | 0.1859 | 0.7879 | 0.3008 |
| | lncRNA | exon | 0.0322 | 0.2018 | 0.0556 | 0.0581 | 0.2794 | 0.0961 | 0.0363 | 0.1223 | 0.0560 | 0.0434 | 0.2722 | 0.0748 |
| | all RNA | exon | 0.2338 | 0.5619 | 0.3302 | 0.2928 | 0.6035 | 0.3943 | 0.2788 | 0.6347 | 0.3874 | 0.4453 | 0.7700 | 0.5643 |
| Caduceus PS | mRNA | exon | 0.1222 | 0.6024 | 0.2032 | 0.2044 | 0.6326 | 0.3089 | 0.2562 | 0.5887 | 0.3571 | 0.1536 | 0.6750 | 0.2503 |
| | | CDS | 0.1576 | 0.7123 | 0.2581 | 0.3206 | 0.7460 | 0.4485 | 0.3459 | 0.7244 | 0.4682 | 0.2118 | 0.8027 | 0.3352 |
| | lncRNA | exon | 0.0359 | 0.2268 | 0.0619 | 0.0429 | 0.2452 | 0.0730 | 0.0418 | 0.1249 | 0.0626 | 0.0370 | 0.2387 | 0.0640 |
| | all RNA | exon | 0.3298 | 0.6429 | 0.4360 | 0.3566 | 0.6608 | 0.4632 | 0.4290 | 0.6831 | 0.5270 | 0.4903 | 0.7937 | 0.6062 |

| Sequnce length | | | 4096 | | | 4096 | | | 4096 | | |
|---|---|---|---|---|---|---|---|---|---|---|---|---|
| Gene type | | | all | | | all | | | all | | |
| Species | | | human | | | human | | | human | | |
| Test-time augmentation | | | rev comp | | | splice site filter | | | rev comp&splice site filter | | |
| Model/dataset | | | precision | recall | f1 | precision | recall | f1 | precision | recall | f1 |
| GENA base | mRNA | exon | 0.0381 | 0.4774 | 0.0706 | 0.2426 | 0.4109 | 0.3051 | 0.2298 | 0.4770 | 0.3102 |
| | | CDS | 0.0409 | 0.5206 | 0.0759 | 0.2656 | 0.4450 | 0.3327 | 0.2526 | 0.5200 | 0.3400 |
| | lncRNA | exon | 0.0279 | 0.3300 | 0.0514 | 0.1677 | 0.2945 | 0.2137 | 0.1548 | 0.3300 | 0.2108 |
| | all RNA | exon | 0.1477 | 0.2712 | 0.1912 | 0.3708 | 0.1809 | 0.2432 | 0.4550 | 0.2607 | 0.3315 |
| Caduceus PH | mRNA | exon | 0.1687 | 0.6251 | 0.2657 | 0.6143 | 0.5583 | 0.5849 | 0.6897 | 0.6241 | 0.6553 |
| | | CDS | 0.2296 | 0.7375 | 0.3502 | 0.6940 | 0.6626 | 0.6780 | 0.7768 | 0.7361 | 0.7559 |
| | lncRNA | exon | 0.0448 | 0.2413 | 0.0755 | 0.2684 | 0.2018 | 0.2304 | 0.3180 | 0.2413 | 0.2744 |
| | all RNA | exon | 0.3355 | 0.6035 | 0.4313 | 0.6383 | 0.5572 | 0.5950 | 0.6890 | 0.6017 | 0.6424 |
| Caduceus PS | mRNA | exon | 0.1604 | 0.6467 | 0.2570 | 0.6408 | 0.6010 | 0.6203 | 0.6795 | 0.6454 | 0.6620 |
| | | CDS | 0.2174 | 0.7614 | 0.3383 | 0.7272 | 0.7105 | 0.7188 | 0.7692 | 0.7596 | 0.7644 |
| | lncRNA | exon | 0.0436 | 0.2551 | 0.0745 | 0.2821 | 0.2268 | 0.2515 | 0.3106 | 0.2551 | 0.2801 |
| | all RNA | exon | 0.3781 | 0.6668 | 0.4826 | 0.7285 | 0.6392 | 0.6809 | 0.7472 | 0.6633 | 0.7028 |

Table A12: PR AUC benchmark, related to Figure 1.

| | Caduceus PS | GENA large | SegmentNT | SegmentNT multi species | SegmentBorzoi | SegmentEnformer |
|---|---|---|---|---|---|---|
| Mean | 0.6799 | 0.6348 | 0.6110 | 0.6095 | 0.5329 | 0.5200 |
| 5UTR | 0.5173 | 0.5003 | 0.3752 | 0.3721 | 0.1910 | 0.1914 |
| Exon | 0.9545 | 0.9493 | 0.7674 | 0.7683 | 0.6954 | 0.6755 |
| Intron | 0.9360 | 0.9296 | 0.8421 | 0.8396 | 0.8391 | 0.8382 |
| 3UTR | 0.5425 | 0.5312 | 0.4594 | 0.4581 | 0.4060 | 0.3749 |
| CDS | 0.4492 | 0.2637 | - | - | - | - |

Table A13: BUSCO completeness computed on hold-out gene set (human chromosome 20). Related to A13

| Model | BUSCO dataset | Class | Complete | Fragmented | Ground truth Complete | Ground truth Fragmented |
|---|---|---|---|---|---|---|
| Caduceus PS | Mammalia | EXON | 210 | 36 | 275 | 3 |
| | | CDS | 215 | 33 | 275 | 3 |
| | Primates | EXON | 322 | 40 | 409 | 4 |
| | | CDS | 323 | 41 | 409 | 4 |
| GENA large | Mammalia | EXON | 206 | 35 | 275 | 3 |
| | | CDS | 209 | 39 | 275 | 3 |
| | Primates | EXON | 300 | 48 | 409 | 4 |
| | | CDS | 307 | 49 | 409 | 4 |
| SegmentNT | Mammalia | EXON | 166 | 46 | 275 | 3 |
| | | CDS | 169 | 43 | 275 | 3 |
| | Primates | EXON | 237 | 60 | 409 | 4 |
| | | CDS | 247 | 58 | 409 | 4 |
| SegmentNT multi species | Mammalia | EXON | 168 | 48 | 275 | 3 |
| | | CDS | 169 | 48 | 275 | 3 |
| | Primates | EXON | 232 | 70 | 409 | 4 |
| | | CDS | 237 | 70 | 409 | 4 |
| SegmentBorzoi | Mammalia | EXON | 36 | 33 | 275 | 3 |
| | | CDS | 36 | 33 | 275 | 3 |
| | Primates | EXON | 54 | 39 | 409 | 4 |
| | | CDS | 53 | 38 | 409 | 4 |
| SegmentEnformer | Mammalia | EXON | 31 | 27 | 275 | 3 |
| | | CDS | 31 | 31 | 275 | 3 |
| | Primates | EXON | 40 | 28 | 409 | 4 |
| | | CDS | 39 | 28 | 409 | 4 |
| Tiberius | Mammalia | EXON | 232 | 6 | 275 | 3 |
| | | CDS | 232 | 6 | 275 | 3 |
| | Primates | EXON | 347 | 3 | 409 | 4 |
| | | CDS | 347 | 3 | 409 | 4 |
| AUGUSTUS | Mammalia | EXON | 194 | 27 | 275 | 3 |
| | | CDS | 192 | 30 | 275 | 3 |
| | Primates | EXON | 278 | 46 | 409 | 4 |
| | | CDS | 279 | 54 | 409 | 4 |

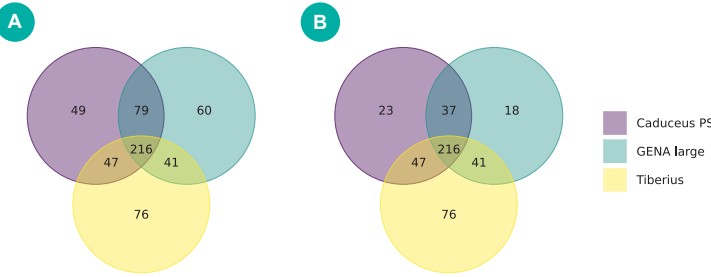

Figure A2: Each model provides unique set of annotated genes, yet large portion of errors are shared accross models. Overlap of correctly segmented genes shown for protein-coding and lncRNA genes together (A), and for protein-coding genes only (B).

Table A14: Exon- and CDS-level benchmark. Related to Figure 1

| Model | Gene type | Class | precision | recall | f1 |
|---|---|---|---|---|---|
| Caduceus PS | mRNA | EXON | 0.9215 | 0.8763 | 0.8983 |
| | | CDS | 0.8928 | 0.8562 | 0.8741 |
| | lncRNA | EXON | 0.5232 | 0.4293 | 0.4717 |
| | all RNA | CDS | 0.8412 | 0.7750 | 0.8068 |
| GENA large | mRNA | EXON | 0.8877 | 0.8778 | 0.8827 |
| | | CDS | 0.8350 | 0.8156 | 0.8252 |
| | lncRNA | EXON | 0.5208 | 0.5174 | 0.5191 |
| | all RNA | CDS | 0.8043 | 0.7962 | 0.8002 |
| SegmentNT | mRNA | EXON | 0.3303 | 0.7554 | 0.4597 |
| | | CDS | 0.0722 | 0.6697 | 0.1304 |
| | lncRNA | EXON | 0.0044 | 0.0797 | 0.0084 |
| | all RNA | CDS | 0.1030 | 0.6025 | 0.1760 |
| SegmentNT multi species | mRNA | EXON | 0.1893 | 0.7577 | 0.3029 |
| | | CDS | 0.0353 | 0.6707 | 0.0671 |
| | lncRNA | EXON | 0.0027 | 0.0889 | 0.0052 |
| | all RNA | CDS | 0.0568 | 0.6064 | 0.1039 |
| SegmentBorzoi | mRNA | EXON | 0.0203 | 0.0647 | 0.0309 |
| | | CDS | 0.0038 | 0.0488 | 0.0070 |
| | lncRNA | EXON | 0.0006 | 0.0039 | 0.0011 |
| | all RNA | CDS | 0.0129 | 0.0509 | 0.0206 |
| SegmentEnformer | mRNA | EXON | 0.0008 | 0.0037 | 0.0013 |
| | | CDS | 0.0002 | 0.0021 | 0.0003 |
| | lncRNA | EXON | 0.0000 | 0.0007 | 0.0000 |
| | all RNA | CDS | 0.0003 | 0.0030 | 0.0006 |
| Tiberius | mRNA | EXON | 0.7484 | 0.5930 | 0.6617 |
| | | CDS | 0.9288 | 0.7880 | 0.8526 |
| | lncRNA | EXON | 0.5439 | 0.0204 | 0.0393 |
| | all RNA | CDS | 0.7456 | 0.4633 | 0.5715 |
| AUGUSTUS | mRNA | EXON | 0.6710 | 0.6018 | 0.6345 |
| | | CDS | 0.7539 | 0.6911 | 0.7211 |
| | lncRNA | EXON | 0.3118 | 0.0381 | 0.0680 |
| | all RNA | CDS | 0.6572 | 0.4742 | 0.5509 |

Table A15: Gene level metrics computed on a gene set assembled from 14 animal species. Metrics are calculated for protein-coding and non-coding genes in a gene set from a single chromosome for each species. MRCA MYA - million years from most recent common ancestor with Homo sapiens.

| Species | MRCA (MYA) | Chromosome | Gene type | Class | Caduceus PS | GENA large | Tiberius | AUGUSTUS | SegmentNT | SegmnetNT multi species | Ground truth |
|---|---|---|---|---|---|---|---|---|---|---|---|
| Anopheles funestus | 686 | NC_064599.1 | mRNA | EXON | 1142 | 1533 | 0 | 44 | 286 | 298 | 4821 |
| | | | | CDS | 639 | 322 | 970 | 1907 | 0 | 0 | |
| | | | lncRNA | EXON | 15 | 27 | 0 | 0 | 0 | 0 | 243 |
| | | | all RNA | EXON | 1157 | 1560 | 0 | 44 | 286 | 298 | 5064 |
| Drosophila melanogaster | 686 | NT_033779.5 | mRNA | EXON | 955 | 1079 | 0 | 135 | 391 | 482 | 2657 |
| | | | | CDS | 661 | 462 | 843 | 1431 | 0 | 0 | |
| | | | lncRNA | EXON | 239 | 256 | 0 | 0 | 87 | 2 | 526 |
| | | | all RNA | EXON | 1194 | 1335 | 0 | 135 | 478 | 484 | 3183 |
| Danio rerio | 429 | NC_007114.7 | mRNA | EXON | 480 | 467 | 5 | 0 | 91 | 185 | 1325 |
| | | | | CDS | 262 | 139 | 420 | 0 | 0 | 0 | |
| | | | lncRNA | EXON | 48 | 91 | 0 | 0 | 3 | 4 | 222 |
| | | | all RNA | EXON | 528 | 558 | 0 | 0 | 94 | 189 | 1547 |
| Mugil cephalus | 429 | NC_061770.1 | mRNA | EXON | 613 | 681 | 0 | 0 | 118 | 166 | 2119 |
| | | | | CDS | 386 | 219 | 775 | 3 | 0 | 0 | |
| | | | lncRNA | EXON | 71 | 92 | 0 | 0 | 7 | 8 | 293 |
| | | | all RNA | EXON | 684 | 773 | 0 | 0 | 125 | 174 | 2412 |
| Paralichthys olivaceus | 429 | NC_091093.1 | mRNA | EXON | 321 | 317 | 0 | 0 | 76 | 109 | 944 |
| | | | | CDS | 222 | 98 | 404 | 0 | 0 | 0 | |
| | | | lncRNA | EXON | 15 | 22 | 0 | 0 | 0 | 0 | 129 |
| | | | all RNA | EXON | 336 | 339 | 0 | 0 | 0 | 0 | 1073 |
| Xenopus laevis | 352 | NC_054386.1 | mRNA | EXON | 426 | 440 | 3 | 10 | 90 | 106 | 1463 |
| | | | | CDS | 248 | 106 | 487 | 164 | 0 | 0 | |
| | | | lncRNA | EXON | 30 | 46 | 0 | 1 | 2 | 2 | 161 |
| | | | all RNA | EXON | 456 | 486 | 0 | 11 | 92 | 108 | 1624 |
| Anas platyrhynchos | 319 | NC_092591.1 | mRNA | EXON | 449 | 429 | 2 | 0 | 124 | 104 | 1002 |
| | | | | CDS | 310 | 166 | 624 | 285 | 0 | 0 | |
| | | | lncRNA | EXON | 34 | 43 | 0 | 0 | 0 | 0 | 412 |
| | | | all RNA | EXON | 483 | 472 | 0 | 0 | 124 | 104 | 1414 |
| Gallus gallus | 319 | NC_052536.1 | mRNA | EXON | 463 | 422 | 0 | 0 | 166 | 116 | 1036 |
| | | | | CDS | 331 | 191 | 614 | 295 | 0 | 0 | |
| | | | lncRNA | EXON | 26 | 34 | 0 | 0 | 1 | 0 | 314 |
| | | | all RNA | EXON | 489 | 456 | 0 | 0 | 167 | 116 | 1350 |
| Taeniopygia guttata | 319 | NC_133030.1 | mRNA | EXON | 472 | 423 | 0 | 0 | 127 | 127 | 976 |
| | | | | CDS | 325 | 173 | 596 | 261 | 0 | 0 | |
| | | | lncRNA | EXON | 31 | 40 | 0 | 0 | 0 | 0 | 245 |
| | | | all RNA | EXON | 503 | 463 | 0 | 0 | 127 | 127 | 1221 |
| Bubalus bubalis | 94 | NC_059174.1 | mRNA | EXON | 658 | 673 | 3 | 17 | 161 | 147 | 1239 |
| | | | | CDS | 482 | 355 | 745 | 240 | 0 | 0 | |
| | | | lncRNA | EXON | 47 | 69 | 0 | 0 | 0 | 1 | 331 |
| | | | all RNA | EXON | 705 | 742 | 0 | 17 | 161 | 148 | 1570 |
| Panthera tigris | 94 | NC_056673.1 | mRNA | EXON | 580 | 627 | 0 | 42 | 223 | 193 | 1136 |
| | | | | CDS | 491 | 402 | 681 | 214 | 0 | 0 | |
| | | | lncRNA | EXON | 36 | 62 | 0 | 0 | 2 | 0 | 284 |
| | | | all RNA | EXON | 616 | 689 | 0 | 42 | 225 | 193 | 1420 |
| Tursiops truncatus | 94 | NC_047043.1 | mRNA | EXON | 486 | 498 | 0 | 20 | 162 | 134 | 1079 |
| | | | | CDS | 380 | 287 | 624 | 194 | 0 | 0 | |
| | | | lncRNA | EXON | 29 | 22 | 0 | 0 | 0 | 2 | 214 |
| | | | all RNA | EXON | 515 | 520 | 0 | 20 | 162 | 136 | 1293 |
| Pan troglodytes | 6.4 | NC_072417.2 | mRNA | EXON | 630 | 692 | 11 | 20 | 208 | 165 | 1304 |
| | | | | CDS | 501 | 371 | 797 | 232 | 0 | 0 | |
| | | | lncRNA | EXON | 37 | 57 | 0 | 0 | 0 | 1 | 284 |
| | | | all RNA | EXON | 667 | 749 | 0 | 20 | 208 | 166 | 1588 |
| Homo sapiens | 0 | NC_060944.1 | mRNA | EXON | 297 | 299 | 0 | 11 | 105 | 97 | 546 |
| | | | | CDS | 204 | 130 | 380 | 106 | 0 | 0 | |
| | | | lncRNA | EXON | 68 | 84 | 0 | 0 | 0 | 0 | 434 |
| | | | all RNA | EXON | 365 | 299 | 0 | 11 | 105 | 97 | 980 |

Table A16: Exon- and CDS-level computed on a gene set assembled from 14 animal species. Metrics are calculated for protein-coding and non-coding genes in a gene set from a single chromosome for each species. MRCA MYA - million years from most recent common ancestor with Homo sapiens.

| Species | MRCA (MYA) | Chromosome | Gene type | Class | Caduceus PS | | | GENA-large | | | Tiberius | | | AUGUSTUS | | | SegmentNT | | | SegmentNT multi species | | |
|---|---|---|---|---|---|---|---|---|---|---|---|---|---|---|---|---|---|---|---|---|---|---|
| | | | | | precision | recall | f1 | precision | recall | f1 | precision | recall | f1 | precision | recall | f1 | precision | recall | f1 | precision | recall | f1 |
| Anopheles funestus | 686 | NC_084599.1 | mRNA | EXON | 0.6512 | 0.6048 | 0.6271 | 0.6086 | 0.5773 | 0.5925 | 0.3557 | 0.1950 | 0.2519 | 0.4647 | 0.4771 | 0.4708 | 0.0276 | 0.3951 | 0.0515 | 0.1277 | 0.5106 | 0.2043 |
| | | | | CDS | 0.5972 | 0.5564 | 0.5761 | 0.4427 | 0.3966 | 0.4184 | 0.5877 | 0.3514 | 0.4398 | 0.7007 | 0.7128 | 0.7067 | 0.0125 | 0.3142 | 0.0240 | 0.0299 | 0.4174 | 0.0558 |
| | | | lncRNA | EXON | 0.1812 | 0.1812 | 0.1812 | 0.1906 | 0.2425 | 0.2134 | 0.0000 | 0.0000 | 0.0000 | 0.0108 | 0.0041 | 0.0059 | 0.0009 | 0.0450 | 0.0018 | 0.0049 | 0.0559 | 0.0090 |
| | | | all RNA | EXON | 0.6354 | 0.5916 | 0.6127 | 0.5912 | 0.5668 | 0.5788 | 0.3542 | 0.1889 | 0.2464 | 0.4593 | 0.4623 | 0.4608 | 0.0249 | 0.3841 | 0.0468 | 0.1173 | 0.4984 | 0.1897 |
| Drosophila melanogaster | 686 | NT_033779.5 | mRNA | EXON | 0.7238 | 0.5576 | 0.6299 | 0.6995 | 0.5550 | 0.6189 | 0.3394 | 0.1852 | 0.2396 | 0.5779 | 0.5320 | 0.5540 | 0.0802 | 0.3988 | 0.1336 | 0.2249 | 0.5986 | 0.3267 |
| | | | | CDS | 0.6361 | 0.5226 | 0.5738 | 0.5231 | 0.4241 | 0.4685 | 0.6380 | 0.3816 | 0.4776 | 0.8173 | 0.7878 | 0.8023 | 0.0246 | 0.2802 | 0.0452 | 0.0420 | 0.4348 | 0.0767 |
| | | | lncRNA | EXON | 0.3861 | 0.4229 | 0.4036 | 0.3935 | 0.4761 | 0.4309 | 0.0000 | 0.0000 | 0.0000 | 0.0000 | 0.0000 | 0.0000 | 0.0218 | 0.1310 | 0.0374 | 0.0069 | 0.0164 | 0.0097 |
| | | | all RNA | EXON | 0.6943 | 0.5491 | 0.6132 | 0.6710 | 0.5500 | 0.6045 | 0.3392 | 0.1735 | 0.2296 | 0.5773 | 0.4985 | 0.5350 | 0.0758 | 0.3819 | 0.1266 | 0.2126 | 0.5600 | 0.3082 |
| Danio rerio | 429 | NC_007114.7 | mRNA | EXON | 0.8806 | 0.8329 | 0.8561 | 0.8303 | 0.7837 | 0.8063 | 0.6890 | 0.3685 | 0.4988 | 0.0000 | 0.0000 | 0.0000 | 0.0392 | 0.6103 | 0.0736 | 0.3293 | 0.7380 | 0.4554 |
| | | | | CDS | 0.8127 | 0.7768 | 0.7943 | 0.7183 | 0.6377 | 0.6756 | 0.8621 | 0.5172 | 0.6455 | 0.7311 | 0.6880 | 0.7089 | 0.0244 | 0.5334 | 0.0466 | 0.0749 | 0.6446 | 0.1342 |
| | | | lncRNA | EXON | 0.6737 | 0.6515 | 0.6624 | 0.7095 | 0.7650 | 0.7362 | 0.1132 | 0.0068 | 0.0128 | 0.0000 | 0.0000 | 0.0000 | 0.0150 | 0.3031 | 0.0285 | 0.0224 | 0.1544 | 0.0392 |
| | | | all RNA | EXON | 0.8657 | 0.8201 | 0.8423 | 0.8207 | 0.7824 | 0.8010 | 0.6844 | 0.3616 | 0.4732 | 0.0000 | 0.0000 | 0.0000 | 0.0370 | 0.5886 | 0.0696 | 0.2712 | 0.6868 | 0.3905 |
| Mugil cephalus | 429 | NC_061770.1 | mRNA | EXON | 0.8712 | 0.8262 | 0.8481 | 0.8453 | 0.8223 | 0.8337 | 0.7291 | 0.4967 | 0.5909 | 0.0000 | 0.0000 | 0.0000 | 0.1519 | 0.6816 | 0.2471 | 0.3735 | 0.7077 | 0.4890 |
| | | | | CDS | 0.8293 | 0.7869 | 0.8075 | 0.7704 | 0.7253 | 0.7472 | 0.8758 | 0.6251 | 0.7295 | 0.7259 | 0.7212 | 0.7235 | 0.0568 | 0.5900 | 0.1037 | 0.0812 | 0.6403 | 0.1442 |
| | | | lncRNA | EXON | 0.6201 | 0.5960 | 0.6078 | 0.6149 | 0.6179 | 0.6164 | 0.1585 | 0.0129 | 0.0239 | 0.0000 | 0.0000 | 0.0000 | 0.0379 | 0.2189 | 0.0646 | 0.0372 | 0.1642 | 0.0607 |
| | | | all RNA | EXON | 0.8602 | 0.8162 | 0.8376 | 0.8350 | 0.8134 | 0.8241 | 0.7260 | 0.4757 | 0.5748 | 0.0000 | 0.0000 | 0.0000 | 0.1454 | 0.6423 | 0.2372 | 0.3414 | 0.6841 | 0.4555 |
| Paralichthys olivaceus | 429 | NC_091080.1 | mRNA | EXON | 0.8550 | 0.8525 | 0.8537 | 0.8462 | 0.8402 | 0.8432 | 0.7825 | 0.5505 | 0.6463 | 0.0000 | 0.0000 | 0.0000 | 0.1310 | 0.7329 | 0.2223 | 0.3459 | 0.7899 | 0.4811 |
| | | | | CDS | 0.8274 | 0.8224 | 0.8249 | 0.7977 | 0.7468 | 0.7725 | 0.8224 | 0.6754 | 0.7798 | 0.7664 | 0.7715 | 0.7689 | 0.0610 | 0.6663 | 0.1117 | 0.0814 | 0.7173 | 0.1463 |
| | | | lncRNA | EXON | 0.3907 | 0.3754 | 0.3829 | 0.4461 | 0.4986 | 0.4709 | 0.0769 | 0.0056 | 0.0104 | 0.0000 | 0.0000 | 0.0000 | 0.0000 | 0.0000 | 0.0000 | 0.0000 | 0.0000 | 0.0000 |
| | | | all RNA | EXON | 0.8411 | 0.8377 | 0.8394 | 0.8322 | 0.8296 | 0.8309 | 0.7802 | 0.5336 | 0.6337 | 0.0000 | 0.0000 | 0.0000 | 0.1310 | 0.7329 | 0.2223 | 0.3459 | 0.7899 | 0.4811 |
| Xenopus laevis | 352 | NC_054386.1 | mRNA | EXON | 0.8416 | 0.8741 | 0.8575 | 0.7843 | 0.8186 | 0.8011 | 0.7265 | 0.5094 | 0.5989 | 0.5731 | 0.5644 | 0.5687 | 0.0658 | 0.6977 | 0.1202 | 0.1498 | 0.7469 | 0.2497 |
| | | | | CDS | 0.7979 | 0.8236 | 0.8105 | 0.7035 | 0.6674 | 0.6850 | 0.8579 | 0.6340 | 0.7292 | 0.6371 | 0.6156 | 0.6261 | 0.0328 | 0.6318 | 0.0624 | 0.0851 | 0.6815 | 0.1188 |
| | | | lncRNA | EXON | 0.5230 | 0.5112 | 0.5171 | 0.5233 | 0.6421 | 0.5767 | 0.0968 | 0.0061 | 0.0115 | 0.1156 | 0.0348 | 0.0535 | 0.0091 | 0.1902 | 0.0173 | 0.0166 | 0.1861 | 0.0305 |
| | | | all RNA | EXON | 0.8316 | 0.8620 | 0.8465 | 0.7741 | 0.8127 | 0.7929 | 0.7246 | 0.4926 | 0.5865 | 0.5683 | 0.5467 | 0.5573 | 0.0622 | 0.6808 | 0.1130 | 0.1402 | 0.7311 | 0.2253 |
| Anas platyrhynchos | 319 | NC_092591.1 | mRNA | EXON | 0.9175 | 0.8986 | 0.9079 | 0.8799 | 0.8836 | 0.8818 | 0.8011 | 0.6421 | 0.7129 | 0.0000 | 0.0000 | 0.0000 | 0.3865 | 0.7668 | 0.5139 | 0.3019 | 0.7676 | 0.4334 |
| | | | | CDS | 0.8929 | 0.8785 | 0.8857 | 0.9265 | 0.8037 | 0.8149 | 0.9420 | 0.7958 | 0.8627 | 0.8309 | 0.7945 | 0.8123 | 0.0779 | 0.7016 | 0.1402 | 0.0850 | 0.7089 | 0.1191 |
| | | | lncRNA | EXON | 0.4154 | 0.3475 | 0.3784 | 0.4189 | 0.4557 | 0.4366 | 0.0476 | 0.0013 | 0.0026 | 0.0000 | 0.0000 | 0.0000 | 0.0015 | 0.0594 | 0.0192 | 0.0047 | 0.0826 | 0.0088 |
| | | | all RNA | EXON | 0.8866 | 0.8343 | 0.8501 | 0.8222 | 0.8337 | 0.8279 | 0.7977 | 0.5674 | 0.6631 | 0.0000 | 0.0000 | 0.0000 | 0.2918 | 0.6841 | 0.4091 | 0.1595 | 0.6877 | 0.2589 |
| Gallus gallus | 319 | NC_052536.1 | mRNA | EXON | 0.9088 | 0.8796 | 0.8939 | 0.8657 | 0.8635 | 0.8646 | 0.7975 | 0.6172 | 0.6959 | 0.0000 | 0.0000 | 0.0000 | 0.4097 | 0.7654 | 0.5337 | 0.2824 | 0.7506 | 0.4104 |
| | | | | CDS | 0.8866 | 0.8595 | 0.8728 | 0.8205 | 0.7973 | 0.8087 | 0.9434 | 0.7702 | 0.8480 | 0.8411 | 0.7809 | 0.8099 | 0.0801 | 0.6924 | 0.1437 | 0.0849 | 0.6900 | 0.1186 |
| | | | lncRNA | EXON | 0.4117 | 0.3923 | 0.4018 | 0.3648 | 0.4515 | 0.4035 | 0.5000 | 0.0077 | 0.0152 | 0.0000 | 0.0000 | 0.0000 | 0.0128 | 0.5644 | 0.0213 | 0.0043 | 0.0961 | 0.0081 |
| | | | all RNA | EXON | 0.8639 | 0.8349 | 0.8492 | 0.8100 | 0.8257 | 0.8178 | 0.7969 | 0.5814 | 0.6587 | 0.0000 | 0.0000 | 0.0000 | 0.3248 | 0.7011 | 0.4439 | 0.1540 | 0.6907 | 0.2518 |
| Taeniopygia guttata | 319 | NC_133030.1 | mRNA | EXON | 0.9184 | 0.8936 | 0.9059 | 0.8885 | 0.8808 | 0.8846 | 0.8041 | 0.6583 | 0.7239 | 0.0000 | 0.0000 | 0.0000 | 0.3236 | 0.7608 | 0.4540 | 0.2872 | 0.7691 | 0.3966 |
| | | | | CDS | 0.8917 | 0.8666 | 0.8789 | 0.8247 | 0.8007 | 0.8126 | 0.9450 | 0.8112 | 0.8734 | 0.8455 | 0.7924 | 0.8181 | 0.0736 | 0.6906 | 0.1330 | 0.0868 | 0.7001 | 0.1220 |
| | | | lncRNA | EXON | 0.4441 | 0.4532 | 0.4486 | 0.3971 | 0.4871 | 0.4375 | 0.0000 | 0.0000 | 0.0000 | 0.0000 | 0.0000 | 0.0000 | 0.0052 | 0.0801 | 0.0097 | 0.0024 | 0.0868 | 0.0046 |
| | | | all RNA | EXON | 0.8878 | 0.8664 | 0.8770 | 0.8514 | 0.8565 | 0.8540 | 0.8032 | 0.6176 | 0.6983 | 0.0000 | 0.0000 | 0.0000 | 0.2273 | 0.7187 | 0.3454 | 0.1460 | 0.7289 | 0.2432 |
| Bubalus bubalis | 94 | NC_059174.1 | mRNA | EXON | 0.8737 | 0.8877 | 0.8806 | 0.8141 | 0.8116 | 0.8129 | 0.7181 | 0.5734 | 0.6377 | 0.6769 | 0.6085 | 0.6398 | 0.1441 | 0.7227 | 0.2403 | 0.1609 | 0.7300 | 0.2637 |
| | | | | CDS | 0.8640 | 0.8421 | 0.8529 | 0.8329 | 0.8007 | 0.8165 | 0.9014 | 0.7636 | 0.8268 | 0.7551 | 0.6826 | 0.7170 | 0.0587 | 0.6336 | 0.1074 | 0.0459 | 0.6379 | 0.0857 |
| | | | lncRNA | EXON | 0.4867 | 0.4427 | 0.4636 | 0.4591 | 0.5141 | 0.4850 | 0.0588 | 0.0010 | 0.0020 | 0.0423 | 0.0091 | 0.0149 | 0.0007 | 0.0504 | 0.0015 | 0.0016 | 0.0846 | 0.0031 |
| | | | all RNA | EXON | 0.8689 | 0.8375 | 0.8519 | 0.8343 | 0.8552 | 0.8446 | 0.7168 | 0.5236 | 0.6052 | 0.6628 | 0.5546 | 0.6039 | 0.0643 | 0.6163 | 0.1153 | 0.0769 | 0.6739 | 0.1381 |
| Panthera tigris | 94 | NC_056673.1 | mRNA | EXON | 0.9273 | 0.8944 | 0.9105 | 0.9160 | 0.9098 | 0.9129 | 0.7896 | 0.6065 | 0.6861 | 0.7597 | 0.6524 | 0.6799 | 0.4394 | 0.7809 | 0.5624 | 0.3425 | 0.7900 | 0.4778 |
| | | | | CDS | 0.9062 | 0.8804 | 0.8931 | 0.8771 | 0.8643 | 0.8707 | 0.9388 | 0.7492 | 0.8334 | 0.7769 | 0.7108 | 0.7424 | 0.1344 | 0.7027 | 0.2256 | 0.0832 | 0.7092 | 0.1489 |
| | | | lncRNA | EXON | 0.4403 | 0.3829 | 0.4135 | 0.5428 | 0.5274 | 0.5350 | 0.0000 | 0.0000 | 0.0000 | 0.0722 | 0.0077 | 0.0138 | 0.0036 | 0.0438 | 0.0067 | 0.0021 | 0.0548 | 0.0041 |
| | | | all RNA | EXON | 0.8973 | 0.8594 | 0.8775 | 0.8903 | 0.8829 | 0.8866 | 0.7887 | 0.5839 | 0.6576 | 0.7042 | 0.6071 | 0.6521 | 0.2914 | 0.7291 | 0.4164 | 0.1863 | 0.7384 | 0.2976 |
| Tursiops truncatus | 94 | NC_047043.1 | mRNA | EXON | 0.8898 | 0.8662 | 0.8827 | 0.8861 | 0.8636 | 0.8646 | 0.7751 | 0.5795 | 0.6631 | 0.5929 | 0.5813 | 0.5870 | 0.2771 | 0.7496 | 0.4047 | 0.2199 | 0.7466 | 0.3400 |
| | | | | CDS | 0.8819 | 0.8478 | 0.8644 | 0.8309 | 0.8047 | 0.8176 | 0.9256 | 0.7250 | 0.8131 | 0.6609 | 0.6399 | 0.6502 | 0.0918 | 0.6760 | 0.1616 | 0.0540 | 0.6802 | 0.1000 |
| | | | lncRNA | EXON | 0.4633 | 0.4134 | 0.4369 | 0.4178 | 0.4461 | 0.4474 | 0.0000 | 0.0000 | 0.0000 | 0.0150 | 0.0240 | 0.0042 | 0.0058 | 0.0587 | 0.0078 | 0.0019 | 0.0509 | 0.0037 |
| | | | all RNA | EXON | 0.8743 | 0.8378 | 0.8556 | 0.8339 | 0.8397 | 0.8368 | 0.7735 | 0.5431 | 0.6382 | 0.5839 | 0.5458 | 0.5642 | 0.2068 | 0.7062 | 0.3200 | 0.1274 | 0.7070 | 0.2159 |
| Pan troglodytes | 6.4 | NC_072417.2 | mRNA | EXON | 0.9043 | 0.8768 | 0.8904 | 0.8909 | 0.8928 | 0.8919 | 0.7797 | 0.6354 | 0.7002 | 0.7128 | 0.6240 | 0.6655 | 0.2530 | 0.7574 | 0.3794 | 0.2199 | 0.7559 | 0.3406 |
| | | | | CDS | 0.8749 | 0.8539 | 0.8643 | 0.8408 | 0.8321 | 0.8364 | 0.9262 | 0.7934 | 0.8555 | 0.7667 | 0.6959 | 0.7386 | 0.0925 | 0.6816 | 0.1630 | 0.0534 | 0.6802 | 0.0980 |
| | | | lncRNA | EXON | 0.4914 | 0.3851 | 0.4318 | 0.5182 | 0.5304 | 0.5242 | 0.0000 | 0.0000 | 0.0000 | 0.0357 | 0.0034 | 0.0062 | 0.0010 | 0.0473 | 0.0019 | 0.0012 | 0.0484 | 0.0023 |
| | | | all RNA | EXON | 0.8825 | 0.8451 | 0.8634 | 0.8664 | 0.8694 | 0.8679 | 0.7793 | 0.5945 | 0.6745 | 0.7078 | 0.5840 | 0.6400 | 0.1186 | 0.7116 | 0.2033 | 0.1209 | 0.7103 | 0.2057 |
| Homo sapiens | 0 | NC_060944.1 | mRNA | EXON | 0.9215 | 0.8763 | 0.8983 | 0.8877 | 0.8778 | 0.8827 | 0.7484 | 0.5930 | 0.6617 | 0.6710 | 0.6018 | 0.6345 | 0.3303 | 0.7554 | 0.4597 | 0.1893 | 0.7577 | 0.3029 |
| | | | | CDS | 0.8928 | 0.8562 | 0.8741 | 0.8350 | 0.8156 | 0.8252 | 0.9288 | 0.7880 | 0.8526 | 0.7539 | 0.6911 | 0.7211 | 0.0722 | 0.6897 | 0.1304 | 0.0353 | 0.6707 | 0.0671 |
| | | | lncRNA | EXON | 0.5232 | 0.4293 | 0.4717 | 0.5208 | 0.5174 | 0.5191 | 0.5439 | 0.0204 | 0.0393 | 0.3118 | 0.0381 | 0.0680 | 0.0044 | 0.0797 | 0.0084 | 0.0027 | 0.0889 | 0.0052 |
| | | | all RNA | EXON | 0.8412 | 0.7750 | 0.8068 | 0.8043 | 0.7962 | 0.8002 | 0.7456 | 0.4633 | 0.5715 | 0.6572 | 0.4742 | 0.5509 | 0.1030 | 0.6025 | 0.1760 | 0.0568 | 0.6064 | 0.1039 |

## APPENDIX H. CLUSTERIZATION OF HIDDEN STATES OF THE MODELS

**Setup**  We extracted final-layer hidden states for ten randomly selected human genes, comprising six mRNA and four lncRNA transcripts. Two model states were analyzed: pretrained HuggingFace (HF) weights and our fine-tuned `GENATATOR` models for both architectures. For `GENA-LM` (BPE tokenization), each token embedding was expanded uniformly across its nucleotide span to obtain one vector per base. Importantly, we intercepted embeddings directly from the RMT backbone prior to the U-NET decoder in order to evaluate the pretrained representation itself. This was necessary because the U-NET component was introduced only in this work and is randomly initialized, as no pretrained version with U-NET exists. Passing embeddings through such a randomly initialized head would risk altering the information contained in the pretrained backbone. For `Caduceus`, weight tying was disabled (`weight_tying=False`) for both HF and fine-tuned states, which doubled the number of trainable parameters (up to 16M parameters). We fit two-dimensional PCA directly to the raw per-base embeddings and then applied $k$-means with $k=5$.

**Homogeneity metric**  Let $K$ denote the ground-truth label random variable over exon, intron, CDS, $5'$UTR, and $3'$UTR, and $C$ the cluster assignment returned by $k$-means. Define

$$H(K) = -\sum_k \frac{n_k}{N} \log\left(\frac{n_k}{N}\right), \qquad H(K \mid C) = -\sum_c \sum_k \frac{n_{c,k}}{N} \log\left(\frac{n_{c,k}}{n_c}\right),$$

where $n_k$ is the count of label $k$, $n_c$ is the size of cluster $c$, $n_{c,k}$ is the number of samples with label $k$ in cluster $c$, and $N$ is the total number of samples. The homogeneity score is

$$\mathrm{h} = 1 - \frac{H(K \mid C)}{H(K)},$$

with h=1 when $H(K)$=0 (see `sklearn.metrics.homogeneity_score`).

**Selected gene set**  The analysis covered the ten human genes listed in Table A17, spanning both coding and non-coding classes and a broad range of transcript lengths.

Table A17: Gene set used for the embedding analysis. Lengths are transcript lengths in base pairs.

| Gene | Type | Length (bp) |
|---|---|---|
| LOC105375876 | lncRNA | 4,791 |
| CPSF1 | mRNA | 16,281 |
| FDFT1 | mRNA | 36,533 |
| OSER1-DT | lncRNA | 14,964 |
| ERGIC3 | mRNA | 15,580 |
| TPX2 | mRNA | 62,507 |
| NOP56 | mRNA | 5,768 |
| IQANK1 | mRNA | 56,563 |
| LINC02986 | lncRNA | 3,453 |
| LOC107986930 | lncRNA | 140,852 |

**Explained variance of PCA**  To evaluate how much variance in the embeddings is captured by the leading principal components, we report the explained variance ratios (EVR) of the first two components (Table A18). These values quantify how strongly base identity or higher-order transcript structure dominate the embedding space.

Table A18: Explained variance ratios (EVR) of the first two PCA components computed directly on per-base embeddings without pooling.

| Model state | $EVR_1$ | $EVR_2$ |
|---|---|---|
| Caduceus PS (HF) | 0.587 | 0.164 |
| Caduceus PS (fine-tuned) | 0.477 | 0.221 |
| GENA LM large (HF) | 0.010 | 0.009 |
| GENA LM large (fine-tuned) | 0.515 | 0.078 |

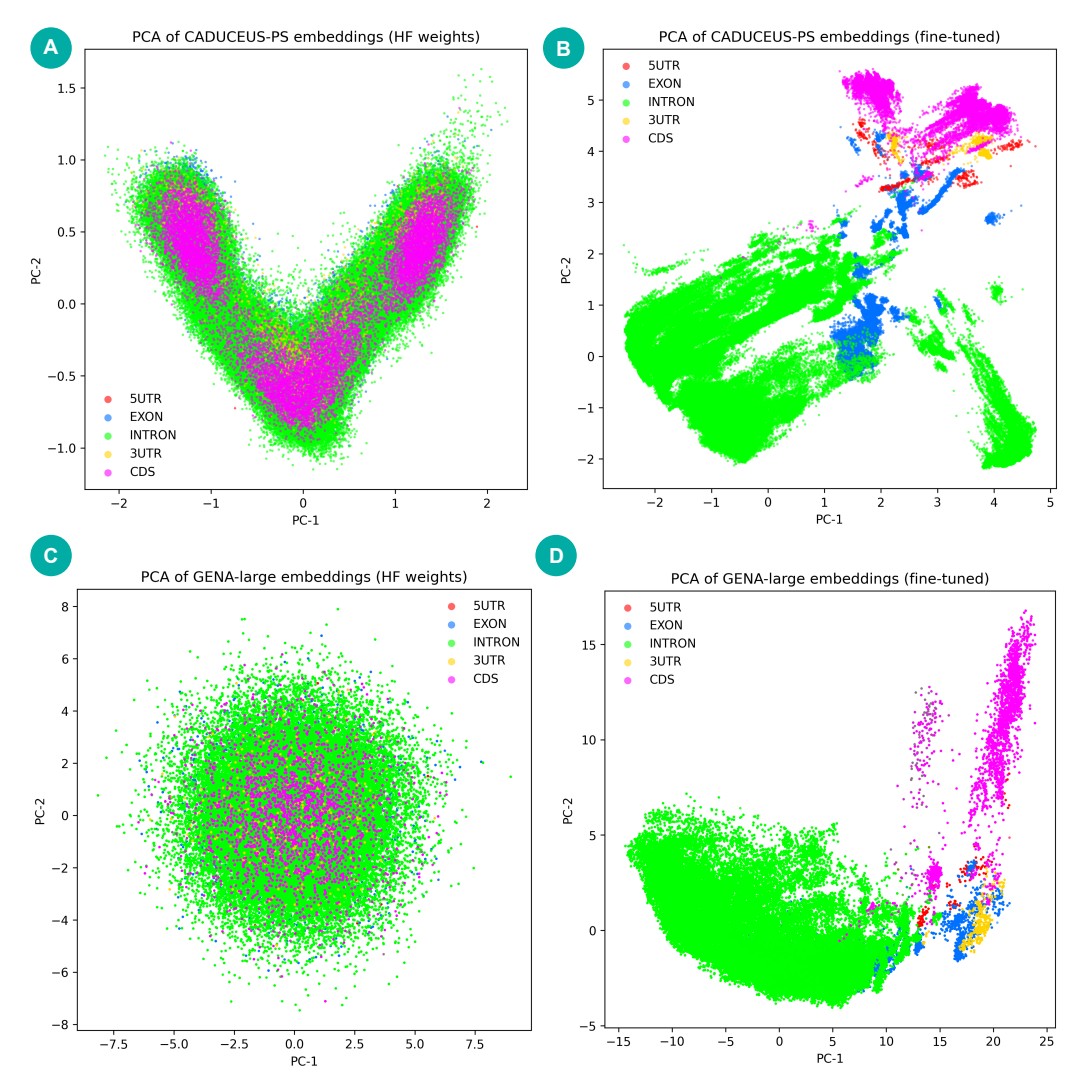

Figure A3: PCA of final-layer embeddings colored by gene-structure labels (5'UTR, EXON, IN-TRON, 3'UTR, CDS). Panels correspond to `Caduceus PS` with HuggingFace (HF) weights (A), `Caduceus PS` after fine-tuning (B), `GENA LM large` with HF weights (C), and `GENA LM large` after fine-tuning (D).

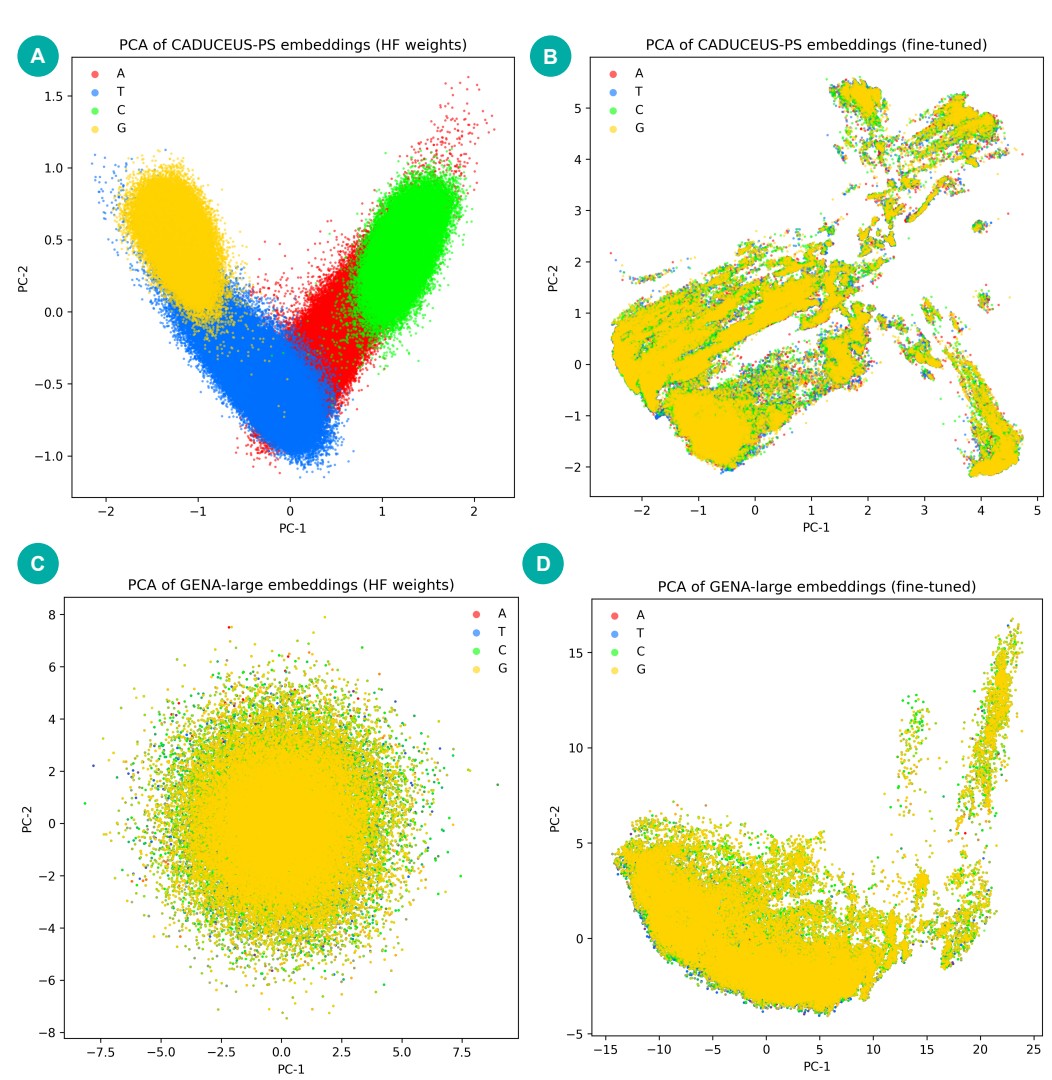

Figure A4: PCA of the same embeddings colored by nucleotide identity (A, T, C, G). Under HF weights, Caduceus PS exhibits clear separation by base identity, while fine-tuning reduces base-driven structure and enhances organization by transcript elements.

APPENDIX I. GENATATOR ERROR ANALYSIS.

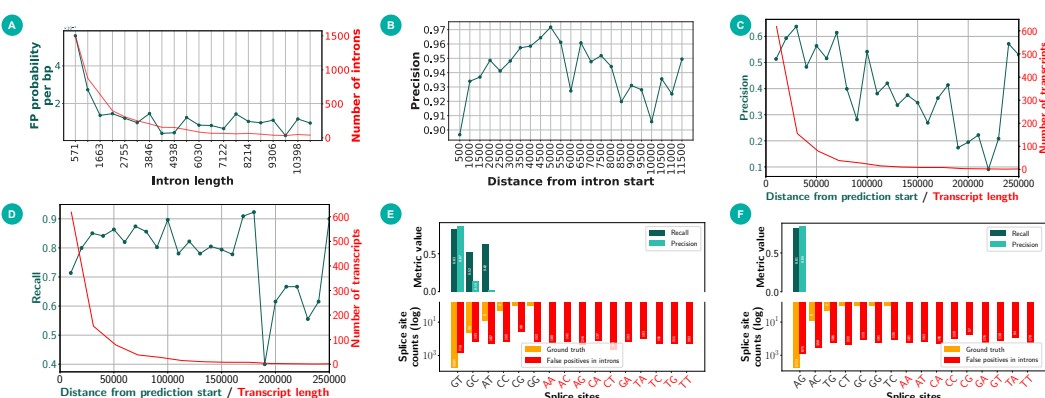

Figure A5: Error analysis provides insights into potential tweaks for improving gene annotation. A-D: Performance metrics as a function of intron length (A), distance from exon-intron boundary (B), and distance from gene sequence start (C-D). B aggregates intron sequences located at specific distance from exon-intron boundary. In A and B the distribution is cropped at the 90th percentile, in C and D at 250Kb. E and F: Precision and recall at predicted intron-exon boundaries, stratified by flanking dinucleotide, separately for left (E) and right (F) intron boundary, with the distribution of targets shown in red and orange.

APPENDIX J. COMPUTING POWER REQUIREMENTS.

We intentionally performed vast majority of the experiments on a small dataset using downscaled models (i.e. base GENA-LM version instead of large) to save computation time and allow more datasets and architectures to be benchmarked. We believe that providing results of the thorough benchmarking is important background with saves compute for others who is going to develop better models for gene annotation.

Average time and resources required for processing 250 Kbp with the most efficient GENATATOR models are provided in the table below. For the whole human chromosome it takes 15 min using single A100 GPU and 8.5 GB GRAM. GENA-based models can be used even without GPU: with Intel(R) Xeon(R) Platinum 8358 CPU @ 2.60GHz single chromosome (chr20, 67Mbp) can be annotated within 3h.

Table A19: Runtime and memory usage of different models.

| Model | A100 × 1 Time | A100 × 1 Memory | CPU Time | CPU Memory |
|---|---|---|---|---|
| GENA large | 3.5 s | 8 430 MiB | 42 s | 8 430 MiB |
| Caduceus PS | 1 s | 7 936 MiB | NA | NA |

Here, NA indicates that Caduceus PS cannot be executed on CPU.

In addition to per-chunk throughput, we also measured end-to-end inference time on a full human chromosome using the same hardware configuration (one NVIDIA A100 80GB). On chromosome 20 of the T2T human genome, the GENA-based GENATATOR model required approximately 16 minutes to complete annotation, while the Caduceus-based GENATATOR variant completed the same task in about 8 minutes. For comparison, SegmentNT (evaluated using its recommended window size of 49,992 bp) required 36 minutes, Tiberius completed annotation in 13 minutes, and AUGUSTUS required 67 minutes.

## APPENDIX K. MODELS SCORING AND BENCHMARKING

### K.1 PROCESSING PREDICTIONS

For all models except Tiberius and AUGUSTUS, each nucleotide was assigned the class with the highest value from the comparison group. The comparison group is specific to each class: for the exon class, it includes exon and intron; for the CDS class, it includes CDS, intron, 5'UTR, and 3'UTR.

### K.2 BENCHMARKING

Predictions were obtained by feeding the model with nucleotide sequences of transcripts (for interval level and BUSCO) or genes (for gene level). SegmentNT is not designed to process very long sequences, so for this model, the gene sequence was split into non-overlapping 50 kb segments. For SegmentBorzoi and SegmentEnformer, the input segment length was set to 196608 nucleotides, as recommended by the authors.

For AlphaGenome, several input sequence lengths are available; here, we used a segment size of 1 Mb. For the segmentation task, the most suitable track, splice_sites, was employed. Exons were defined based on acceptor and donor classes, corresponding to the first and last nucleotide of each exon, respectively. Acceptor-donor pairs were identified in a sliding window from the beginning to the end of the sequence. We evaluated thresholds ranging from 0.1 to 0.9 in increments of 0.1, and for the final results, the best-performing threshold was selected.

It is important to note that SegmentNT can predict only the exon class, so metrics for the CDS class were obtained by subtracting predictions of 5'UTR and 3'UTR from exon predictions. Finally, GENATATORs are capable of predicting both exons and CDS, so for these models, metrics were calculated across all classes for all genes and transcripts.

### K.3 INTERVAL LEVEL METRICS

To evaluate the accuracy of exon prediction for each model, sequences of a single transcript per gene were provided (the transcripts with the maximum total exon length were selected).

### K.4 GENE LEVEL

Each model generated predictions based on the gene sequences. Interval-level (exon or CDS) analysis was then performed, comparing predictions to each known transcript of each gene. If there is a transcript with complete and reciprocal overlap between predicted exons and known exons, the gene was considered to be identified. CDS analysis was performed similarly.

### K.5 BUSCO

Based on the predictions of each model, the nucleotide sequences of the genes were obtained for analysis. After performing the translation operation, the corresponding proteins were obtained and the longest of them was selected. The strand for translation was determined either directly if model outputs it explicitly (Tiberius and AUGUSTUS), or based on the predicted classes 5'UTR and 3'UTR, using the formula: $(FirstU5 - FirstU3) - (LastU5 - LastU3)$, where $FirstU5$ is the cumulative probability of 5'-UTR class preidction in the first 50 bases, $LastU3$ is the cumulative probability of 3'-UTR class prediction in the last 50 bases, and etc. (for other models). For AlphaGenome, the strand corresponding to the gene strand was used. Subsequently, the set of obtained proteins was analyzed using BUSCO.

## APPENDIX L.  HOMOLOGY EXCLUSION EXPERIMENT IN *S. cerevisiae*.

To ensure that performance of GENATATORs in yeast is not attributable to residual homology with mammalian training data, we performed a stringent control. All 766 annotated protein-coding genes from S. cerevisiae chromosome NC_001136.10 were compared to the full proteomes of the 39 mammalian species used during training (1,827,441 proteins in total) using BLASTP (E-value cutoff 1e–05). Every yeast gene with at least one significant hit was excluded, resulting in a filtered set of 270 genes without detectable protein-level similarity to the training data.

We then evaluated gene-level reconstruction accuracy on this filtered set. Results are summarized in Table Sx.

Table A20: Gene-level reconstruction on *S. cerevisiae* genes without detectable protein-level homology to mammals.

| Model | Gene level (%) |
|---|---|
| Caduceus PS | 98.52 |
| GENA large | 92.59 |
| AUGUSTUS | 41.85 |

Even under these stringent conditions, GENATATORs recovered over 250 genes - more than twice the number recovered by AUGUSTUS, which was run with a species-specific HMM profile for *S. cerevisiae*. These findings demonstrate that the observed performance cannot be explained by homology leakage, but instead reflects the models' ability to capture general splice and coding sequence patterns transferable across kingdoms.

## APPENDIX M.  GENATATORs GENERALIZE ACROSS UNSEEN SPECIES.

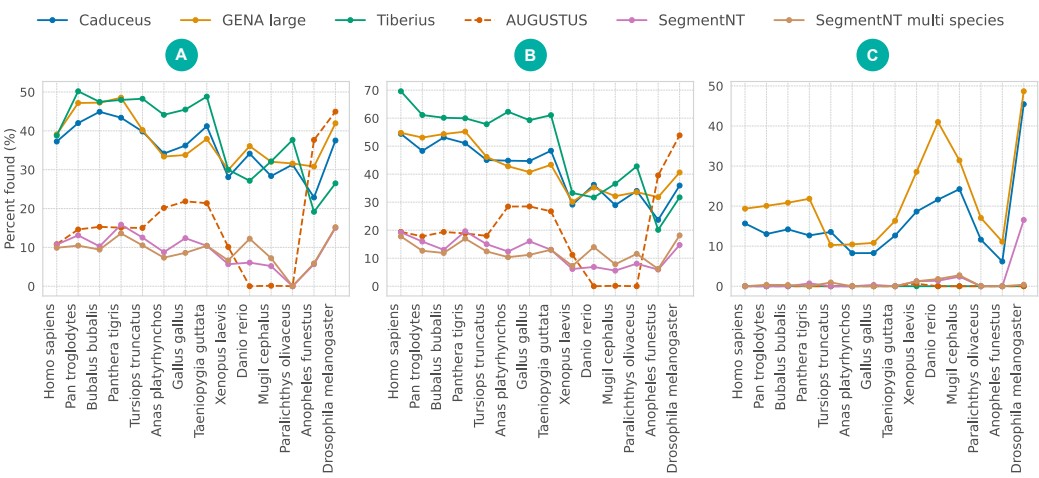

Figure A6: GENATATORs generalize to previously unseen species. Performance of the models in human and 13 other species for all (A), protein-coding (B), and lncRNA (C) genes. See Appendix D Tables A15 and A16 for more information on metrics.

## APPENDIX N.  LIMITATIONS.

While GENATATORs demonstrate strong performance in benchmarking studies, their accuracy remains far from perfect. Currently, only approximately 30–40% of all human genes can be correctly segmented by any of the models evaluated in this study.

Another limitation lies in gene discovery. Although gene segmentation is a critical component of genome annotation, complete annotation also requires accurate identification of gene boundaries, including non-coding untranslated regions (UTRs), which remains challenging for all evaluated tools.

Finally, the poor results observed in our embedding-only training experiments highlight a fundamental limitation of current DNA language models: they do not capture gene structure during the pretraining phase. This underscores the need for architectural or training paradigm improvements in future DNA LM development.

## APPENDIX O. DECLARATION OF LLM USAGE.

Large Language Models (LLMs) were used solely to improve the readability and clarity of the manuscript text. No parts of the analysis, results, or conclusions were generated by LLMs.

