# OpenReview forum: "GENATATORs: ab initio Gene Annotation With DNA Language Models"
_ICLR.cc/2026/Conference — Submitted to ICLR 2026_

### Official Review · Reviewer_qq8z · 2025-10-29

**Soundness:** 1
**Presentation:** 1
**Contribution:** 2
**Rating:** 2
**Confidence:** 5

**Summary:**

This paper presents GENATATORs, an ab initio gene annotation approach built by fine-tuning long-context DNA language models to segment genomic DNA including mRNAs and lncRNAs. The authors argue that token-level metrics are insufficient and introduce interval-level and gene-level criteria to better reflect biological correctness and study architecture and training choices. Results show strong improvements on lncRNA and UTR segmentation and competitive overall gene-level scores.

**Strengths:**

1. GENATATORs tackles a de novo annotation with long-context DNA LMs and emphasizes interval and gene-level metrics aligned with biological correctness.

2. Solid ablations show clear contributions from long context, multi-species training, and RC augmentation.

3. Strong performance on lncRNA and UTR where HMM-style tools underperform and potential for first-pass annotation in poorly annotated species.

**Weaknesses:**

1. Coverage of recent DNA-LM foundations remains incomplete like Evo.

2. The writing style is lacking a smooth logical narrative, and the reading experience is not friendly enough.

3. Beyond stating frameshift risk, quantify downstream protein-level impact when boundaries shift by 1–3 bp and compare to token-level metrics to demonstrate practical delta.

4. Keep claims strictly as validation. For NMD, report total counts of qualifying genes, baseline, and statistical significance.

**Questions:**

Please see Weaknesses

---

> ### Author Response · Authors · 2025-12-03
> **Official Comment by Authors**
>
> We thank the reviewer for the comments.
>
> We note that we include Evo2-1B in the embeddings comparison. While being the largest model in our benchmark, it does not provide any substantial advance over smaller model’s embeddings. Given high costs of fine-tunning such models, we proceed with smaller models in future experiments, which allows us to perform multiple finetunning experiments. Benchmarking even larger Evo2-40B  model is not practical given the intention to build a whole-genome annotation pipeline, since inference time and cost for the complete genomes will be too high for end-users.
>
> As for the writing style, we simplified the narrative. The intention is to make the paper easier to follow for both ML and biology audiences.
>
> For the gap between token-level metrics and actual recovery of full transcripts, we refer directly to Appendix C and Table A5. When context increases from roughly 32k to 250k nucleotides with BPE outputs, mean PR-AUC changes only slightly, but the number of fully reconstructed genes grows from 44 to 106. When keeping the same 250k context and switching from BPE outputs to nucleotide outputs, PR-AUC increases modestly, but the number of reconstructed genes grows by another 102. Together with the idea in Appendix C, this shows that context length and base-level precision have a large effect on whether complete transcripts are recovered, even when token-level scores look similar. This is why interval- and gene-level metrics are central in our paper.
>
> Finally, regarding NMD, we did not include a specific analysis in the submission. We believe that the referee has confused our submission with some other work.

---

### Official Review · Reviewer_gzrs · 2025-11-01

**Soundness:** 3
**Presentation:** 3
**Contribution:** 2
**Rating:** 4
**Confidence:** 4

**Summary:**

This paper introduces a family of fine-tuned DNA language models on the gene annotation task. The authors critique existing evaluation metrics, and propose a new interval-level metric to show that fine-tuned DNA language models outperform classical HMM-based models and SegmentNT. The models also generalize to evolutionarily distant species, indicating cross-species generalization.

**Strengths:**

1. The paper aims to tackle a fundamental task for genomics, and the ability to annotate non-coding genes and UTRs shows real progress compared to previous works.

2. The paper designs novel and rigorous evaluation metrics, the interval and gene-level metrics seem capable of capturing biologically meaningful aspects of gene segmentation performance.

3. The models demonstrate strong cross-species generalization, showing the potential of fine-tuned DNA language models to capture evolutionarily conserved sequence patterns across diverse genomes.

**Weaknesses:**

1. Even though the authors show that increasing the input context improves model performance, the paper lacks an analysis of performance across different gene lengths, which would clarify whether the models exhibit bias toward genes of certain lengths or whether the longer input context specifically helps in learning longer or more complex genes.

2. In the cross-species generalization task, it seems that the best models are different for different species. The paper lacks an analysis of this phenomenon. In general, it would be better for the authors to provide guidance for readers on how to choose which type(s) of DNA-LM to use in different scenarios.

3. The work’s main novelty resides in the proposed evaluation metrics and systematic benchmarking; however, the modeling methodology largely follows existing DNA LM fine-tuning frameworks, making the technical contribution relatively weak.

**Questions:**

1. Is there any specific reason to choose chr 8, 20, 21 for validation/test?

---

> ### Author Response · Authors · 2025-12-03
> **Official Comment by Authors**
>
> Thanks for the suggestions that can help strengthen our paper.
>
> You asked whether the models show a bias toward genes of certain lengths and whether longer context specifically helps long or complex genes. We computed performance across bins of transcript lengths. The updated supplementary Figure A5 in updated Appendix I explains that precision and recall remain steady through most of the transcript and begin to drop only at the extreme tail where very few transcripts reach maximal length. This shows that increasing context helps recover long genes without harming the short ones, and we do not observe a systematic bias toward any particular length.
>
> For cross-species use, we provide a simple recommendation. When the goal is high-accuracy CDS recovery in vertebrate protein-coding genes, Tiberius remains a strong choice. When the goal includes UTRs and lncRNA or when working with non-vertebrate genomes, GENATATOR with long context performs better in our benchmarks. These choices follow naturally from the design focus of each model and are consistent with our BUSCO and gene-level comparisons.
>
> You also asked how we chose chromosomes for validation and test, and whether this makes the comparison across tools fair. GENATATOR is trained on 39 mammalian genomes, including human, but for human we exclude chromosomes 8, 20 and 21 from training. SegmentNT, in its original work, is trained on human with chromosomes 20 and 21 used as the held-out set. In our benchmark we evaluate both GENATATOR and SegmentNT on chromosome 20, which is unseen for GENATATOR and belongs to the official held-out set for SegmentNT, so there is no leakage for either model. Tiberius is trained on a panel of mammalian species and is not fitted on human data, so evaluation on human chromosome 20 is also fully out-of-training for Tiberius. This choice of chromosome 20 as the main test region therefore gives a common, unseen human target for all three tools, despite their different training regimes.
>
> We appreciate your concern about methodological novelty and agree that our paper does not propose a fundamentally new model architecture or introduce substantial algorithmic innovations. However, we would like to highlight that originality is not strictly limited to the development of entirely new methods or algorithms. Specifically, contributions providing novel insights by carefully evaluating existing approaches, or demonstrating significant improvements in aspects such as efficiency, fairness, or accuracy, can be considered equally valuable. Our main intent was not to propose yet another new model, but rather to rigorously evaluate the existing state-of-the-art DNA language models, such as GENA-LM and Caduceus, in the specific and biologically critical context of gene segmentation, with a particular focus on systematic evaluation across the entire annotation pipeline rather than isolated subtasks. Furthermore, our experiments uncovered significant insights, notably demonstrating the superior capability of fine-tuned DNA language models in annotating long non-coding RNA (lncRNA) genes, which prior methods, including state-of-the-art tools such as Tiberius, consistently miss. Given the biological importance of lncRNAs, we consider this finding particularly valuable for biological research and for those who develop DNA LM models. Additionally, we conducted extensive cross-species evaluations, explicitly demonstrating substantially improved annotation transferability to evolutionarily distant genomes. We believe these outcomes form a rigorous analytical foundation upon which future architectural innovations might be built, clearly identifying and quantifying the factors that critically influence DNA LM-based genome annotation.

---

### Official Review · Reviewer_LjS9 · 2025-11-01

**Soundness:** 2
**Presentation:** 3
**Contribution:** 2
**Rating:** 4
**Confidence:** 4

**Summary:**

The paper presents strategies for large scale gene annotation. The authors introduce a new interval-based evaluation metric, and provide theoretical justification that this more reliably captures annotation quality. They proceed by fine-tuning various pre-trained DNA language models, investigating the role of modeling design choices such as architecture and training procedures. Finally, they show that fine-tuned models outperform existing gene annotation tools, in particular in an out-of-domain setting.

**Strengths:**

**Originality** The authors present a novel segmentation scoring procedure, which appears to better capture DNA annotation quality than current metrics.

**Quality** The introduced metric appears well thought through, and the fine-tuning benchmarks are comprehensive, allowing them to draw  conclusions on best practices in the field.

**Clarity** The paper is well-written and easy to follow.

**Significance** The paper presents a convincing case for the potential of DNA LMs for gene annotation, which is likely to have impact on the community.

**Weaknesses:**

The paper seems to want to do several things at once: The title suggests a new prediction method (GENATATORS), but the focus in the paper itself is split primarily between discussion of a new segmentation metric, and a fine-tuning benchmark study. Any of these would be interesting, but maybe not all for the ICLR audience. The authors should consider whether they can give the paper a clearer focus, and whether ICLR is the right venue for it (compared to e.g. a Bioinformatics journal)

Since the segmentation metric is central to the paper, it would be helpful if the authors could make it clear how the presented method differs from that presented in Scalzitti et al 2020, which they cite as inspiration. It would also be helpful to the reader if the score could be contrasted clearly to the gene-level score used in Tiberius. If there is sufficient history here, it might even make sense to dedicate a pargraph in Related Work to this.

I have some concerns regarding the claim made on line 376, related to Fig 1. The authors state. *"Specifically, the GENA-based GENATATOR marginally outperforms Tiberius, while the Caduceus-based variant performs slightly below it."*. As far as I remember (correct me if I'm wrong), Tiberius is not designed to predict lncRNA, so it seems unfair to compare it against GENA that does, and then lumping together the results in both categories. As far as I can see, on the mRNA case, Tiberius clearly outperforms GENA, and it is therefore not correct that GENA "performs on par with the current -state-of-the-art model".

In my opinion, another weakness of the paper is the almost aggresive stance towards earlier work in this field. The BEND benchmark is descibed as "not biologically rigorous", and Tiberius is claimed to "consistently fail" for lncRNA, although it was not trained on this task (as far as I remember). In science, we continuously seek to improve over earlier work, but it should be sufficient to highlight own merits rather than describe earlier work in derogatory terms.

**Questions:**

## Questions

line 164. *"To account for this ambiguity, we use a gene level rule that accepts a prediction as correct when the predicted interval set exactly matches the interval set of any annotated isoform of the target gene."*. Is this generally reliable? How complete is our ground truth annotation of isoforms?

line 224. *"none of the models produced embeddings containing sufficient information for accurate gene segmentation"*. I was a bit puzzled about this conclusion. As far as I remember, reasonable performances have been reported for gene annotation from fixed embeddings in the past - e.g. in BEND. You state *"These observations are consistent with recent findings reported in the Nucleotide Transformer embeddings"*. Which results are you referring to here?

line 236. *"Together, these results indicate that pretraining alone is insufficient to encode the features required for precise gene segmentation and that task-specific fine-tuning remains essential for achieving high segmentation accuracy."*. Are you sure that your results warrant this strong conclusion? It might be valid for the simplistic linear mapping that you use here, but earlier results show that better performance can be obtained by more structured decoding strategies, such as the one presented here: https://arxiv.org/abs/2505.03377. It is also not unlikely that future DNA LMs will incorporate gene structure in the pretraining, and the statement you make should therefore at least be qualified to *current* language models.

line 269. *"We also found that using multiple isoforms per gene slightly reduced accuracy, confirming that the single-isoform strategy remains preferable"*. Is it a general recommendation that we should ignore multiple isoform information. Does it not instead suggest that the model you are using is not rich enough?


## Minor details

The references to figures seems unnecessarily elaborate ("Appendix A Figure A1 B"). For clarity, consider shortening them.

---

> ### Author Response · Authors · 2025-12-03
> **Official Comment by Authors (part 1)**
>
> Thanks for the helpful suggestions.
>
> Speaking about metrics, we evaluate predictions at the interval level and at the gene level. An exon or CDS is counted as correct only when the predicted start and end exactly match the reference at base resolution. A gene is counted as correct only when the full set of predicted exons exactly matches the exon set of at least one annotated isoform of that gene. This allows joint evaluation of mRNA and lncRNA and measures whether the full structure is reconstructed, not just local boundaries.
>
> Scalzitti et al. report nucleotide-level sensitivity and specificity and a protein-based score for canonical protein-coding genes in their G3PO benchmark. The dataset is built from 1,793 UniProt proteins, and evaluation is carried out only for protein-coding genes. As a result, the benchmark does not include lncRNA at all and does not evaluate complete gene structure at the gene level. Also, each gene is represented by a single reference protein rather than by all of its annotated transcript structures.
>
> Tiberius, introduced by Gabriel et al., uses a network with a differentiable HMM to predict ab initio protein-coding gene structure. For each gene they retain only the transcript with the longest coding sequence as the reference, and when comparing to pipelines that output multiple isoforms they again keep only the longest coding isoform per gene. Therefore exon- and gene-level metrics in Tiberius are always computed against one coding isoform rather than across the full isoform set, in contrast to our evaluation, which accepts any annotated isoform as correct and is more biologically aligned with how real genes are organized.
>
> Our interval-level and gene-level metrics operate on the full exon structure of each transcript. When we evaluate exons, we include every exon type: UTR exons, coding exons, and exons belonging to lncRNAs. This means that any valid transcript structure, whether it contains only coding exons, only non-coding exons, or a combination of UTR and coding segments, must be reconstructed exactly for the prediction to be counted as correct. At the gene level, the predicted set of exons must match the exon set of at least one annotated isoform. Together, these rules create a more rigorous and biologically aligned evaluation, because even modest boundary shifts in UTRs or non-coding exons can change the interpretation of the whole structure. In parallel, we provide CDS-based metrics, which are similar to Scalzitti et al., allowing users to comprehensively benchmark all properties of models. This approach therefore measures complete gene reconstruction for both coding and non-coding genes, rather than focusing only on recovery of the protein-coding portion.
>
> We included this detailed comparison with Scalzitti et al and Gabriel et al. into our revised submission.
>
> We also acknowledge the reviewer’s important point that the current set of annotated isoforms may be incomplete. Motivated by this, we analyzed false-positive predictions and found that a fraction overlapped so-called poison exons, which are absent from standard references. This suggests that model performance may, in fact, be underestimated. Nonetheless, we observe that increasing model and dataset scale consistently improves performance under our current evaluation protocol (Table 1). This supports the use of reference isoforms as a practical and effective ground truth for model comparison. While imperfect, this remains the most viable approach until experimental techniques enable a more comprehensive and reliable catalog of human transcript isoforms.
>
> Regarding frozen embeddings, we note that in BEND benchmark authors use complex trainable tasks-specific heads on the top , while we process embeddings with linear layers only. Our experiments thus demonstrate what current pretrained encoders capture. In our experiments a linear probe on frozen representations is far below the accuracy of fine-tuning. This gap shows that today’s encoders do not yet encode fully recoverable gene structure, even if a heavier decoder could improve results. We agree that reference to NT paper does not provide a good example of such experiment setup and reworded subsection “Training on embeddings” to clearly articulate our goals.
>
> We thank the reviewer for this valuable clarification regarding the design scope of Tiberius. Indeed, it was not originally developed to identify lncRNA genes, and we revised the manuscript (especially the subsection “Benchmarking GENATATOR against other gene-annotation tools”) to make this distinction explicit. At the same time, we note that from the perspective of end-users seeking comprehensive gene annotation, limitations in lncRNA detection—regardless of origin—can impact practical utility. Our comparison aims to reflect such real-world applicability.

---

> ### Author Response · Authors · 2025-12-03
> **Official Comment by Authors (part 2)**
>
> We also fully agree with the reviewer that scientific progress is best supported through constructive, respectful engagement with prior work. We regret if any phrasing in the manuscript gave the impression of dismissiveness. This was not our intention. Our aim was to critically assess where existing tools and benchmarks may fall short in capturing biological complexity, and to motivate why additional methods and evaluation frameworks are needed. We revised wording throughout the manuscript—for example, softened “consistently fails” to a more neutral, descriptive formulation—and ensured all comparisons are framed in a collegial and objective manner. We are grateful for this reminder and already made these improvements accordingly.

---

### Official Review · Reviewer_rLYT · 2025-11-02

**Soundness:** 3
**Presentation:** 3
**Contribution:** 2
**Rating:** 4
**Confidence:** 3

**Summary:**

The manuscript introduces GENATATORs, a method for ab initio gene annotation using DNA language models (LMs), designed to segment gene structures such as exons, introns, and UTRs in mRNA and lncRNAs. The method aims to address challenges in accurately annotating gene structures and identifying non-coding genes (e.g., lncRNAs) that are often missed by traditional tools. GENATATOR achieves this by training DNA LMs on long genomic sequences (up to 250 kbp) and leveraging novel biologically grounded evaluation metrics that measure gene segmentation accuracy at the interval and gene levels. The method is evaluated on cross-species benchmarks, comparing its performance to existing tools like AUGUSTUS and SegmentNT, showing superior gene recovery and generalization to new species. GENATATOR excels in segmenting mRNA and lncRNAs, but its limited biological completeness (e.g., lack of cis-regulatory elements and alternative splicing prediction) and reliance on a single isoform per gene are key drawbacks.

**Strengths:**

### Cross-Species Generalization

A major strength of GENATATOR is its ability to generalize across species, with demonstrated strong performance on unseen species, including plants and animals. This cross-species generalization is particularly important for genome annotation, as it can be applied to less-characterized species without the need for species-specific tuning.

**Weaknesses:**

### Limited Biological Completeness

While GENATATOR performs well in annotating mRNA and lncRNA, it is biologically incomplete because it fails to predict key regulatory regions such as promoters, enhancers, and silencers. These cis-regulatory elements are essential for understanding gene regulation, as they control when and how genes are turned on or off in different cell types. Additionally, small non-coding RNAs (such as miRNAs, snoRNAs, and snRNAs) are not included in the predictions, which limits the model’s utility in complete genome annotation. This is a major drawback for using GENATATOR in comprehensive genome-wide annotation tasks where understanding gene regulation is critical.

### Alternative Splicing and Isoform Prediction

The single-isoform assumption is another significant limitation. GENATATOR currently predicts only one canonical isoform for each gene, ignoring alternative splicing and isoform diversity, which are essential for understanding gene function and cell-type-specific gene regulation. Many genes undergo alternative splicing to produce different protein isoforms or regulatory RNAs, and GENATATOR’s static model fails to capture this complexity. Incorporating the ability to predict multiple isoforms for each gene would make the model far more biologically accurate and useful in functional genomics.

### Static Gene Model Assumption

GENATATOR assumes a fixed, static gene structure, which does not account for dynamic gene expression across different cell types or tissues. Gene regulation is highly tissue-specific, and many genes are expressed differently depending on the cell type. By not accounting for these cell-type-specific variations, GENATATOR fails to model the functional complexity of genes, which is crucial for understanding disease mechanisms and gene regulation in different contexts.

**Questions:**

See Weaknesses.

In addition, I'm curious to learn more about the runtime performance of the proposed method versus existing methods like SegmentNT.

---

> ### Author Response · Authors · 2025-12-03
> **Official Comment by Authors (part 1)**
>
> Thanks for the thoughtful and detailed feedback.
>
> Our goal in this work is ab initio gene annotation on a reference genome. Specifically, we focus on segmenting gene bodies into exons, introns and UTRs for mRNA and lncRNA directly from DNA sequence. We fully agree that cis‑regulatory elements such as promoters, enhancers and silencers are essential for understanding gene regulation. However, in practice these elements are defined by cell type context and experimental assay, and different datasets often disagree on their exact boundaries. A given locus can behave as an enhancer in one tissue and be inactive in another. Even considering one tissue and based on the same set of experimental techniques, there is often no consensus about how to define enhancer and promoter (doi: 10.1101/gad.309351.117). Because of this strong context dependence, and because these elements are not part of the gene body itself, we treat regulatory element prediction as a separate problem. The long‑context architecture used in GENATATOR can be reused for segmentation of regulatory regions, but that would require carefully curated, cell‑type‑specific labels and is outside the scope of the present paper.
>
> On the question of small non-coding RNAs, we agree that a complete genome annotation should also include miRNAs, snoRNAs and snRNAs. In this work we restrict the segmentation task to mRNA and lncRNA for two reasons. First, both mRNA and lncRNA are polymerase II transcripts and are consistently annotated across genomes, which makes them suitable for the supervised training setup we use. Other gene classes have a different biological nature: many miRNAs are produced as short hairpin-derived polymerase II precursors, snoRNAs often reside inside introns of host genes and undergo additional processing, and snRNAs span two transcriptional systems, with U1, U2, U4 and U5 transcribed by polymerase II and U6 by polymerase III. Second, these small RNA families are far less abundant. In the T2T human reference there are 20097 mRNAs and 16373 lncRNA, but only 1889 miRNAs, 1190 snoRNAs and 146 snRNAs. Since we expect limited generalizability between mRNA/lncRNA and other RNA classes (due to the difference of mechanisms described above), this imbalance makes it difficult to train and evaluate a robust model for these gene types using the same segmentation framework. We therefore view small RNAs as an important future extension, but focus first on achieving high-quality segmentation for mRNA and lncRNA, which remains a challenging problem on its own.
>
> Alternative splicing are indeed crucial for gene annotation, and we fully agree that this is a natural next step. In the current paper we work with encoder‑only DNA LMs and segmentation heads, which by design produce one set of labels for each input sequence. Many loci have several biologically valid isoforms, so the same DNA can map to different exon–intron structures. When we tried to train GENATATOR on all annotated isoforms per gene, this led to identical inputs with conflicting labels and resulted in a measurable drop in segmentation accuracy (Table 1 in our paper). There are isoforms that share the same genomic span but differ in exon usage, and others that differ in length, and a single deterministic segmentation head is not a good fit for that setting. To predict full isoform sets one would need a generative decoder that can sample multiple valid transcript structures for the same locus, for example a diffusion‑style sampler on top of the encoder. We view this as a logical continuation of our work, but also as a harder, still largely open problem. At the same time, we note that even predicting one isoform with base‑level accuracy is not fully solved yet, and our results focus on pushing that frontier.
>
> You also point out that GENATATOR assumes a static gene model and does not account for cell‑type‑specific expression. We agree that gene regulation is tissue specific and that dynamic usage of promoters, enhancers and splice sites is highly relevant for disease. Here we deliberately stay within the classical ab initio setting, where the input is only the reference DNA sequence and the target is a reference gene annotation. This matches how widely used tools such as AUGUSTUS and Tiberius are evaluated. Models that incorporate chromatin state, expression data or predicted regulatory activity, such as AlphaGenome‑like approaches, are complementary and address a different question, namely where and when a gene is active. GENATATOR is intended as a sequence‑only annotator of possible gene structures. In future work it would be interesting to combine our long‑context segmentation with tissue‑specific regulatory predictors, but that combination is beyond the scope of the current study.

---

> ### Author Response · Authors · 2025-12-03
> **Official Comment by Authors (part 2)**
>
> Finally, regarding runtime, we measured inference time on chromosome 20 of the T2T human genome on the same hardware (one Nvidia A100 80GB). The GENA‑based GENATATOR variant takes 16 minutes, the Caduceus‑based GENATATOR takes 8 minutes, SegmentNT (run with 49992 bp windows) takes 36 minutes, Tiberius takes 13 minutes and AUGUSTUS takes 67 minutes. We added those numbers to Appendix J so readers can now see how GENATATOR compares to existing tools in terms of practical throughput.

---

### Author Response · Authors · 2025-12-03
**Official Comment by Authors to the AC**

Here we provide a concise summary of our rebuttal for the Area Chair, focusing on the main themes raised across all reviews and how we addressed them in our revised paper. Reviewers consistently highlighted the strengths of our work, including strong cross-species generalization, accurate reconstruction of complete gene structures (including UTRs and lncRNAs), comprehensive benchmarking of DNA language models and classical tools, and the introduction of biologically grounded interval- and gene-level metrics suitable for long-context genomic models. There were no high-level concerns related to the study relevance, methodology, results, and conclusions. Referee comments are primarily guidelines about how to polish manuscript style or inquire to provide more details, which we followed in the majority of cases. There were also suggestions how current work can be extended in the future for more complex tasks such as multi-isoform prediction and/or detection of different classes of genomic elements. While we agree that our work opens possibilities for these future applications, we explain that datasets, baselines, and benchmarks included in the manuscript do not allow us to follow all these suggestions within the scope of current work.

In summary, we believe that our work fits well within the ICLR scope. It provides insights into long-context DNA language models, introduces biologically rigorous evaluation metrics and offers systematic evidence on how architectural choices influence structured genomic prediction. These contributions are directly relevant for ML researchers developing foundation models for genomics. We have updated our manuscript PDF to reflect all clarifications and additions described above. We thank all reviewers for their constructive feedback and hope that our revisions address their concerns.

---

### Meta-Review · Area_Chair_QLGW · 2026-01-07

**Summary:**

This paper introduces GENATATORs, an ab initio gene annotation framework based on long-context DNA language models, together with interval- and gene-level evaluation metrics that better reflect biological correctness. Across reviews, there was broad agreement that the work is technically sound and carefully executed, with clear strengths in cross-species generalization and improved recovery of UTRs and lncRNAs compared to existing tools.

The main concerns were mostly about scope and positioning rather than correctness. Reviewer rLYT questioned the biological completeness, noting the lack of regulatory elements and isoform diversity. Reviewer LjS9 and gzrs felt that the paper tries to do several things at once, splitting focus between a new evaluation metric and a large benchmarking study, which makes the core contribution less sharp. Reviewer qq8z was more critical of the presentation and viewed the modeling contribution as incremental, despite acknowledging strong empirical results on long-context modeling.

The rebuttal was very strong and addressed most concerns directly. The authors clearly justified the scope choices (e.g., single-isoform setting), added runtime comparisons, clarified how the proposed metrics differ from prior work, and substantially improved the framing and tone of the paper. Many reviewer concerns were satisfactorily resolved.

That said, the overall review scores remain on the lower side, and some residual concerns about focus and perceived contribution remain. While the work is solid and valuable, overturning the aggregate sentiment would be difficult at this point. So, I lean toward reject this time but strongly encourage authors to resubmit after sharpening the main contribution and narrative to the future venue.

**Reviewer Concerns:**

rLYT

Concerns about biological scope (regulatory elements, small RNAs, isoforms) were clearly addressed by clarifying the ab initio setting and scope limitations. Runtime comparisons were added.

LjS9

Questions about evaluation metrics, comparison to prior benchmarks, and tone toward earlier work were carefully clarified and revised.

gzrs

Analysis by gene length, cross-species model choice, and chromosome split rationale were fully addressed with new experiments and explanations.

qq8z

rLYTWriting clarity and interpretation of interval/gene-level metrics were improved, with added quantitative evidence supporting the claims.

**Reviewer Scores:**

rLYT / LjS9 / gzrs would have slightly increased their scores, but probably not significantly enough to overturn the final decision.

---

### Decision · Program_Chairs · 2026-01-26

Reject